# Real-Valued Backpropagation is Unsuitable for Complex-Valued Neural Networks

**Zhi-Hao Tan, Yi Xie, Yuan Jiang, Zhi-Hua Zhou**
National Key Laboratory for Novel Software Technology,
Nanjing University, Nanjing 210023, China
`{tanzh, xiey, jiangy, zhouzh}@lamda.nju.edu.cn`

## Abstract

Recently complex-valued neural networks have received increasing attention due to successful applications in various tasks and the potential advantages of better theoretical properties and richer representational capacity. However, the training dynamics of complex networks compared to real networks remains an open problem. In this paper, we investigate the dynamics of deep complex networks during real-valued backpropagation in the infinite-width limit via neural tangent kernel (NTK). We first extend the Tensor Program to the complex domain, to show that the dynamics of any basic complex network architecture is governed by its NTK under real-valued backpropagation. Then we propose a way to investigate the comparison of training dynamics between complex and real networks by studying their NTKs. As a result, we surprisingly prove that for most complex activation functions, the commonly used real-valued backpropagation reduces the training dynamics of complex networks to that of ordinary real networks as the widths tend to infinity, thus eliminating the characteristics of complex-valued neural networks. Finally, the experiments validate our theoretical findings numerically.

## 1 Introduction

Recently complex-valued neural networks have been successfully applied to various tasks, such as time-series prediction [Wisdom et al., 2016], computer vision [Trabelsi et al., 2018], signal processing [Yao et al., 2020]. Compared to real-valued neural networks, it is shown that complex networks have the potential to provide richer representational capacity [Arjovsky et al., 2016], faster learning [Danihelka et al., 2016], better motivation and generalization for signal-related tasks [Hirose and Yoshida, 2012, Tygert et al., 2016]. Theoretically, there have been significant advances for complex networks regarding the universal approximation property [Voigtlaender, 2020], critical points [Nitta, 2002], local minima [Nitta, 2013] and separation results [Zhang et al., 2022].

However, training deep complex networks has been challenging because of several non-intuitive analytical properties of complex algebra. Firstly, in practice, we often deal with a real-valued cost function, which is non-analytic with respect to complex-valued parameters. Secondly, Liouville's theorem asserts that every bounded and complex-differentiable function is a constant. Thus almost all activation functions are non-analytic due to the preference for boundedness before the popularity of ReLU [Scardapane et al., 2018]. Moreover, Voigtlaender [2020] has proved that the universal approximation property holds only when using non-holomorphic activation functions.

Due to these reasons, different backpropagation algorithms in the complex domain were independently proposed for non-holomorphic networks [Bassey et al., 2021], mostly by optimizing the real and imaginary components separately. Recently, real-valued backpropagation [Nitta, 1997] is widely used due to the convenience of utilizing the real-valued deep learning library [Arjovsky et al., 2016, Trabelsi et al., 2018, Tan et al., 2020]. It optimized the complex network just like a real network by

36th Conference on Neural Information Processing Systems (NeurIPS 2022).

computing partial derivatives of the cost with respect to the real and imaginary parts separately and achieved state-of-the-art performance.

As a result, a natural and fundamental problem in complex-valued neural networks arises: Do complex networks have a different inductive bias from real networks during training? Do complex networks trained by gradient descent tend to learn different hypotheses from real networks? However, to the best of the authors' knowledge, the training dynamics of backpropagation for complex networks compared to real networks remains open.

In this paper, we investigate the training dynamics of deep complex networks under real-valued backpropagation from neural tangent kernel (NTK) perspective [Jacot et al., 2018], which captures the optimization behavior of neural networks in the infinite-width limit. Then we provide a way to investigate the comparison of training dynamics between complex and real networks via their NTKs. As a result, we obtain informative results which may guide the algorithm selection for complex networks in practice if people want to take full advantage of complex networks.

**Our contributions.** Our main contributions can be summarized as follows:

- First, we extend the Tensor Program [Yang, 2020] to the complex domain and show that for a complex-valued neural network of any basic architecture in the infinite-width limit, the training dynamics of real-valued backpropagation is determined by kernel gradient descent with a deterministic NTK at initialization.

- Second, we investigate the comparison of training dynamics between complex and real networks based on their NTKs. We surprisingly prove that the commonly used real-valued backpropagation reduces the training dynamics of complex-valued multi-layer perceptrons (MLPs) to that of ordinary real MLPs as the widths tend to infinity, thus eliminating the characteristics of complex-valued neural networks. This result holds for most commonly used complex activation functions, including all split activation functions such as $\mathbb{C}$ReLU, $\mathbb{C}$Sigmoid, $\mathbb{C}$tanh; and part of magnitude-based holomorphic activation functions.

- Finally, we study the results numerically, and the experiments verifies our findings. Specifically, in several settings with different depths and various activation functions, the NTKs of complex networks converge to the NTKs of real networks as the widths grow.

**Organization.** We start with some preliminaries and notations about complex networks and neural tangent kernels in Section 2. In Section 3, we investigate the NTK of complex-valued neural networks of any architecture during real-valued backpropagation in the infinite-width limit. Section 4 firstly presents the NTK of complex MLPs in any depth and then investigates the conditions that training dynamics of complex-valued MLPs reduce to that of ordinary real MLPs. We verify our results empirically in Section 5. Finally we discuss the related works and conclude the paper. Due to the limited space, all proofs are placed in the appendices.

## 2 Preliminaries

### 2.1 Complex-Valued Neural Networks

Without loss of generality, we focus on complex-valued neural networks with real-valued output $f_\theta(z) \in \mathbb{R}^{d_{out}}$ with parameter set $\theta \in \mathbb{C}^p$, input $z \in \mathbb{C}^d$ and $z = x + yi$ with $x, y \in \mathbb{R}^d$, and trained by real-valued backpropagation algorithm (also called complex-BP or generalized complex BP, see Appendix A for more details), which is conventional in the literature [Arjovsky et al., 2016, Wisdom et al., 2016, Trabelsi et al., 2018, Zhang and Zhou, 2021, Wu et al., 2021, Zhang et al., 2022]. Our analysis can be naturally applied to complex-valued output by decomposing the real and imaginary part of output into two functions, or applied to real-valued input by treating its imaginary part as zero.

For an $L$-hidden layer complex network, we denote the output of last hidden layer as $\boldsymbol{h}_L \in \mathbb{C}^n$. Without loss of generality, we consider that the output of a complex network with a linear readout layer is achieved via

$$f_\theta(z) = \Re\{\mathbf{W}_{L+1}\boldsymbol{h}_L\}$$

where $\mathbf{W}_{L+1} \in \mathbb{C}^{n \times d_{out}}$ [Wisdom et al., 2016, Zhang et al., 2022]. Note that there are other two common forms of linear readout layer to generate real-valued output, like $f_\theta(z) = \mathbf{W}_{L+1}\Re\{\boldsymbol{h}_L\}$

where the output weight $\mathbf{W}_{L+1} \in \mathbb{R}^{n \times d_{out}}$ [Wu et al., 2021], and $f_\theta(\boldsymbol{z}) = \mathbf{W}_{L+1} \begin{bmatrix} \Re\left(\boldsymbol{h}_L\right) \\ \Im\left(\boldsymbol{h}_L\right) \end{bmatrix}$ where $\mathbf{W}_{L+1} \in \mathbb{R}^{2n \times d_{out}}$ [Arjovsky et al., 2016]. They can be treated as special cases of our settings.

For a complex-valued neural network $f_\theta(\boldsymbol{z})$, we can always decompose all the complex operations into two-dimensional real-valued operations, denoted as $\tilde{f}_{[\theta_R, \theta_I]}([\boldsymbol{x}, \boldsymbol{y}])$ where $\theta_R, \theta_I \in \mathbb{R}^p$ are the real and imaginary parts of all complex parameters respectively. However, it would be erroneous to assume that a complex network is equivalent to an ordinary real-valued neural network, because the operation of complex multiplication limits the degree of freedom [Hirose and Yoshida, 2012].

## 2.2 Neural Tangent Kernel

For a real-valued deep neural network $f_{\theta_r}(\boldsymbol{x}) \in \mathbb{R}^{d_{out}}$ with parameter set $\theta_r \in \mathbb{R}^p$ and input $\boldsymbol{x} \in \mathbb{R}^d$, its Neural Tangent Kernel (NTK) under gradient descent is defined as

$$\widehat{\Theta}_r\left(\boldsymbol{x}, \boldsymbol{x}'\right) = \langle \nabla_{\theta_r} f_{\theta_r}(\boldsymbol{x}), \nabla_{\theta_r} f_{\theta_r}\left(\boldsymbol{x}'\right) \rangle \tag{1}$$

which quantifies the functional gradient descent when taking an infinitely small gradient step on a new observation. In case $f_{\theta_r}$ corresponds to an infinite width MLP, Jacot et al. [2018] showed that $\widehat{\Theta}_r\left(\boldsymbol{x}, \boldsymbol{x}'\right)$ converges to a limiting kernel $\mathring{\Theta}_r\left(\boldsymbol{x}, \boldsymbol{x}'\right)$ at initialization and remains frozen during training, i.e.,

$$\lim_{n \to \infty} \widehat{\Theta}_r^t\left(\boldsymbol{x}, \boldsymbol{x}'\right) = \lim_{n \to \infty} \widehat{\Theta}_r^0\left(\boldsymbol{x}, \boldsymbol{x}'\right) = \mathring{\Theta}_r\left(\boldsymbol{x}, \boldsymbol{x}'\right) \quad \forall \text{ training time } t,$$

which could give an accurate description of training dynamics with kernel gradient descent trajectory. Thus a infinitely wide neural network is governed by a linear model based on its first order Taylor expansion in the parameter space [Lee et al., 2019].

**Tensor Programs.** After the original NTK was derived from multi-layer perceptrons, it is soon extended to a variety of network structures including convolution neural networks (CNTK) [Arora et al., 2019], recurrent neural networks (RNTK) [Yang, 2019, Alemohammad et al., 2020], graph neural networks (GNTK) [Du et al., 2019] and so on. Importantly, Yang [2020], Yang and Littwin [2021] propose NETSOR$^\top$ program, a basic form in Tensor Programs series, and prove that for a real-valued neural network of any architecture that can be represented by NETSOR$^\top$ program language, its NTK converges to a deterministic limit and stays frozen during training in the infinite-width limit.

**Neural Tangent Kernel of complex networks.** For a complex-valued neural network $f_\theta(\boldsymbol{z})$, also denoted as $\tilde{f}_{[\theta_R, \theta_I]}([\boldsymbol{x}, \boldsymbol{y}])$, when it is trained by real-valued backpropagation, the empirical neural tangent kernel is as follows due to that the real and imaginary parts are optimized separately

$$\widehat{\Theta}\left(\boldsymbol{z}, \boldsymbol{z}'\right) = \langle \nabla_\theta f_\theta(\boldsymbol{z}), \nabla_\theta f_\theta\left(\boldsymbol{z}'\right) \rangle = \langle \nabla_{\theta_R} f_{\theta_R}(\boldsymbol{z}), \nabla_{\theta_R} f_{\theta_R}\left(\boldsymbol{z}'\right) \rangle + \langle \nabla_{\theta_I} f_{\theta_I}(\boldsymbol{z}), \nabla_{\theta_I} f_{\theta_I}\left(\boldsymbol{z}'\right) \rangle.$$

We can always rewrite the NTK as $\Theta([\boldsymbol{x}, \boldsymbol{y}], [\boldsymbol{x}', \boldsymbol{y}'])$. Note that at each layer of complex networks in both feed-forward and backward procedure, complex matrix multiplication structure leads to numerous interactions and weight sharing between the real and imaginary parts, which makes the analysis of the complex NTK challenging.

# 3 Complex Tensor Program

In this section, we show that complex-valued neural networks of any basic architecture also have NTK behavior in the infinite-width limit: the training dynamics of real-valued backpropagation is determined by kernel gradient descent with its NTK at initialization. Specifically, we extend the simplified NETSOR$^\top$ program [Yang, 2020] to the complex domain, and propose the basic Complex Tensor Program (CTP). Note that the complex networks are mostly non-holomorphic, thus we do not require the complex tensor program to represent the backward propagation of the complex networks.

**Definition 1 (Complex Tensor Program)** *Given an initial set $\mathcal{V}$ of random $\mathbb{C}^n$ vectors and a set $\mathcal{W}$ of random $\mathbb{C}^{n \times n}$ complex matrices, a sequence of $\mathbb{C}^n$ vectors is called a complex tensor program if they are recursively generated through one of the following ways:*

**ComNonlin** *Given $\phi : \mathbb{C}^k \to \mathbb{C}$ and $\boldsymbol{z}^1, \dots, \boldsymbol{z}^k \in \mathbb{C}^n$, generate $\phi(\boldsymbol{z}^1, \dots, \boldsymbol{z}^k) \in \mathbb{C}^n$;*

**ComMatMul** *Given $\mathbf{W} \in \mathbb{C}^{n \times n}$ and $\boldsymbol{z} \in \mathbb{C}^n$, generate $\mathbf{W}\boldsymbol{z} \in \mathbb{C}^n$.*

Obviously, the complex tensor program could represent the forward procedures of all basic complex network architectures, such as the generic feed-forward full-connected complex networks [Nitta, 2004], the complex-valued recurrent neural networks [Wisdom et al., 2016] and complex-valued convolutional neural networks [Trabelsi et al., 2018, Tan et al., 2020].

## 3.1 Complex network setup

This subsection introduces the settings and assumptions of complex networks considered. Firstly we introduce the required assumption for activation functions of complex networks, which generalizes the assumption in Yang [2020] to the complex domain.

**Assumption 2** *We assume that the complex activation function $\phi : \mathbb{C}^k \to \mathbb{C}$ used in the complex networks and its derivative are polynomially-bounded, i.e., $\phi(\boldsymbol{z})$ satisfies that $|\phi(\boldsymbol{z})| \leq C\|\boldsymbol{z}\|^p + c$ for some $C, p, c > 0$ and $\boldsymbol{z} \in \mathbb{C}^k$; so is its derivative.*

It is worth mentioning that numerous activation functions have been proposed to deal with complex-valued representations and basically they satisfy the assumption. For example, the most commonly used complex sigmoid function $\mathbb{C}$Sigmoid [Nitta, 1997, 2004] and complex hyperbolic tangent function $\mathbb{C}$tanh [Nitta, 2002], which apply sigmoid and hyperbolic tangent function respectively to the real part and imaginary part separately; Moreover, the recently proposed ReLU-based complex activation functions like $\mathbb{C}$ReLU [Trabelsi et al., 2018, Tan et al., 2020], zReLU [Guberman, 2016] and modReLU [Arjovsky et al., 2016] also satisfy the assumption.

**Complex NTK parametrization.** For the complex weights, we initialize each $\mathbf{W} \in \mathcal{W}$ with $\mathbf{W} = \mathbf{A} + \mathbf{B}i = \frac{\sigma_A}{\sqrt{n}} A + \frac{\sigma_B}{\sqrt{n}} Bi$ where $A_{\alpha\beta}, B_{\alpha\beta} \sim \mathcal{N}(0, 1)$, which we refer to as *complex NTK parametrization*. Without loss of generality, we set the variances of real and imaginary parts of all layers as $\sigma_A$ and $\sigma_B$ respectively in the following paper. It naturally extends the real-valued NTK parametrization [Jacot et al., 2018, Lee et al., 2019] to complex parameters. Note that this parametrization is non-vacuous because many previous works [Nitta, 1997, 2004] choose to initialize the real and imaginary parts separately.

**Setup.** Consider a complex-valued neural network $f_\theta(\boldsymbol{z})$ with complex NTK parametrization, its feed-forward procedure can be represented by a complex tensor program and complex activation functions all satisfy Assumption 2. Suppose that there is a multivariate Gaussian $\mathcal{N}_\mathcal{V}$ defined on $\mathbb{R}^{2|\mathcal{V}|}$ such that the real and imaginary variables of the initial set of vectors $\mathcal{V}$ are sampled like $\{\Re[q]_\alpha : q \in \mathcal{V}\} \cup \{\Im[q]_\alpha : q \in \mathcal{V}\} \sim \mathcal{N}_\mathcal{V}$ i.i.d. for each coordinate $\alpha \in [n]$. For the output readout matrix $W_{L+1}$, we also adopt complex NTK parametrization, and it is sampled independently from all other parameters and is not used anywhere else in the interior of the network. Without loss of generality, the network is trained by SGD with batch-size 1 and learning rate 1.

## 3.2 NTK for any complex network

**Theorem 3 (Complex NTK at initialization)** *Consider a complex-valued neural network $f_\theta(\boldsymbol{z})$ with above setup, then as its widths go to infinity, its NTK $\widehat{\Theta}(\boldsymbol{z}, \boldsymbol{z}')$ at initialization converges almost surely to a deterministic limiting kernel $\overset{\circ}{\Theta}(\boldsymbol{z}, \boldsymbol{z}')$ over any finite set of inputs.*

**Corollary 4 (Complex NTK during training)** *Consider training a complex-valued neural network $f_\theta(\boldsymbol{z})$ with above setup. At training time $t$, denote the input sample as $\boldsymbol{z}_t$ and the loss function as $\mathcal{L}_t : \mathbb{R} \to \mathbb{R}$. Suppose $\mathcal{L}_t$ is continuous for all $t$. Then as widths approach infinity, for any $\boldsymbol{z} \in \mathbb{C}^d$ and training time $t$, $f_t(\boldsymbol{z})$ converges almost surely to a random variable $\overset{\circ}{f}_t(\boldsymbol{z})$ and*

$$\overset{\circ}{f}_{t+1}(\boldsymbol{z}) - \overset{\circ}{f}_t(\boldsymbol{z}) = -\overset{\circ}{\Theta}(\boldsymbol{z}, \boldsymbol{z}_t) \, \mathcal{L}'_t\left(\overset{\circ}{f}_t(\boldsymbol{z}_t)\right), \tag{2}$$

*where $\overset{\circ}{\Theta}(\boldsymbol{z}, \boldsymbol{z}_t)$ is the limiting NTK of the complex network at initialization.*

The proofs of Theorem 3 and Corollary 4 are given in Appendix B. Theorem 3 and Corollary 4 show that for a complex-valued neural network of any architecture trained by real-valued backpropagation, its NTK at initialization converges to a deterministic limiting kernel in the infinite-width limit, and the training dynamics is determined by kernel gradient descent with the NTK at initialization.

# 4 Comparison of training dynamics between complex and real networks

In this section, we focus on the important problem: when will complex networks have different inductive bias during training from real networks? The problem can not be solved unless the training dynamics of complex networks could be captured. Based on the NTK theory obtained in Section 3, we could provide a way to compare the training dynamics of complex and real networks by comparing their NTKs. Specifically, we have derived the NTK formula of complex multi-layer perceptrons (MLPs) and investigated the conditions under which complex MLPs trained by real-valued backpropagation will reduce to ordinary real MLPs in the infinite-width limit.

## 4.1 Neural Tangent Kernel of complex multi-layer perceptrons

This subsection presents the NTK formula of the most generic complex network, i.e., the $L$-hidden layer complex MLPs.

The network performs the following computation at layer $l \in [1, L]$

$$\boldsymbol{h}_l = \boldsymbol{s}_l + \boldsymbol{r}_l i = \phi(\mathbf{W}_l \boldsymbol{h}_{l-1}) = \phi\left((\mathbf{A}_l + \mathbf{B}_l i)(\boldsymbol{s}_{l-1} + \boldsymbol{r}_{l-1} i)\right), \tag{3}$$

where $\mathbf{W}_l = \mathbf{A}_l + \mathbf{B}_l i$ is the complex weight matrix and $\boldsymbol{h}_l = \boldsymbol{s}_l + \boldsymbol{r}_l i$ with $\boldsymbol{h}_l \in \mathbb{C}^n$ is the output of $l$-th layer. For the first layer, we set $\boldsymbol{s}_0 = \boldsymbol{x}$ and $\boldsymbol{r}_0 = \boldsymbol{y}$ where $\boldsymbol{z} = \boldsymbol{x} + \boldsymbol{y}i$ and $\boldsymbol{z} \in \mathbb{C}^d$. Besides, the output of the complex network with a linear read-out layer is achieved via $f_\theta(\boldsymbol{x}) = \text{Re}\{\mathbf{W}_{L+1} \boldsymbol{h}_L\}$ where $\mathbf{W}_{L+1} \in \mathbb{C}^{n \times d_{out}}$. Suppose all activation functions $\phi$ satisfy the Assumption 2. Complex NTK parametrization is applied for all complex parameters $\mathbf{W}_1 \in \mathbb{C}^{d \times n}, \mathbf{W}_{L+1} \in \mathbb{C}^{n \times d_{out}}$ and $\mathbf{W}_l \in \mathbb{C}^{n \times n}$ for $l \in [2, L]$.

We denote the real part of the $l$-th hidden layer pre-activation as $\boldsymbol{\alpha}_l(\boldsymbol{z})$ and the imaginary part as $\boldsymbol{\beta}_l(\boldsymbol{z})$. In the feed-forward procedure, we denote the covariance kernel functions between the real and imaginary part of the pre-activations respectively as

$$\Sigma_\alpha^l(\boldsymbol{z}, \boldsymbol{z}') = \mathop{\mathbb{E}}_{\theta \sim \mathcal{N}}\left[\boldsymbol{\alpha}_l(\boldsymbol{z})^\top \boldsymbol{\alpha}_l(\boldsymbol{z}')/n\right], \qquad \Sigma_{\alpha,\beta}^l(\boldsymbol{z}, \boldsymbol{z}') = \mathop{\mathbb{E}}_{\theta \sim \mathcal{N}}\left[\boldsymbol{\alpha}_l(\boldsymbol{z})^\top \boldsymbol{\beta}_l(\boldsymbol{z}')/n\right], \tag{4}$$

$$\Sigma_\beta^l(\boldsymbol{z}, \boldsymbol{z}') = \mathop{\mathbb{E}}_{\theta \sim \mathcal{N}}\left[\boldsymbol{\beta}_l(\boldsymbol{z})^\top \boldsymbol{\beta}_l(\boldsymbol{z}')/n\right], \qquad \Sigma_{\beta,\alpha}^l(\boldsymbol{z}, \boldsymbol{z}') = \mathop{\mathbb{E}}_{\theta \sim \mathcal{N}}\left[\boldsymbol{\beta}_l(\boldsymbol{z})^\top \boldsymbol{\alpha}_l(\boldsymbol{z}')/n\right]. \tag{5}$$

In the backward procedure, we denote the gradient vector of the real part of the $l$-th hidden layer pre-activation as $\boldsymbol{\delta}_\alpha^l(\boldsymbol{z}) := \sqrt{n}\left(\nabla_{\boldsymbol{\alpha}_l(\boldsymbol{z})} f_\theta(\boldsymbol{z})\right)$ and that of the imaginary part as $\boldsymbol{\delta}_\beta^l(\boldsymbol{z}) := \sqrt{n}\left(\nabla_{\boldsymbol{\beta}_l(\boldsymbol{z})} f_\theta(\boldsymbol{z})\right)$. Similarly, we denote the covariance kernel functions of the gradient vector between the real and imaginary part of the pre-activations respectively as

$$\Pi_\alpha^l(\boldsymbol{z}, \boldsymbol{z}') = \mathop{\mathbb{E}}_{\theta \sim \mathcal{N}}\left[\boldsymbol{\delta}_\alpha^l(\boldsymbol{z})^\top \boldsymbol{\delta}_\alpha^l(\boldsymbol{z}')/n\right], \qquad \Pi_{\alpha,\beta}^l(\boldsymbol{z}, \boldsymbol{z}') = \mathop{\mathbb{E}}_{\theta \sim \mathcal{N}}\left[\boldsymbol{\delta}_\alpha^l(\boldsymbol{z})^\top \boldsymbol{\delta}_\beta^l(\boldsymbol{z}')/n\right], \tag{6}$$

$$\Pi_\beta^l(\boldsymbol{z}, \boldsymbol{z}') = \mathop{\mathbb{E}}_{\theta \sim \mathcal{N}}\left[\boldsymbol{\delta}_\beta^l(\boldsymbol{z})^\top \boldsymbol{\delta}_\beta^l(\boldsymbol{z}')/n\right], \qquad \Pi_{\beta,\alpha}^l(\boldsymbol{z}, \boldsymbol{z}') = \mathop{\mathbb{E}}_{\theta \sim \mathcal{N}}\left[\boldsymbol{\delta}_\beta^l(\boldsymbol{z})^\top \boldsymbol{\delta}_\alpha^l(\boldsymbol{z}')/n\right]. \tag{7}$$

**Theorem 5** *For a L-hidden layer complex MLP, with all activation functions satisfying Assumption 2, in the limit as all widths $n \to \infty$, the empirical NTK at initialization converges to the following limiting kernel*

$$\lim_{n \to \infty} \widehat{\Theta}(\boldsymbol{z}, \boldsymbol{z}') = \mathring{\Theta}(\boldsymbol{z}, \boldsymbol{z}') = \Theta(\boldsymbol{z}, \boldsymbol{z}') \otimes \mathbf{I}_{d_{out}} \tag{8}$$

*where*

$$\Theta(\boldsymbol{z}, \boldsymbol{z}') = \sum_{l=1}^{L}\left(\Pi_\alpha^l(\boldsymbol{z}, \boldsymbol{z}')\Sigma_\alpha^l(\boldsymbol{z}, \boldsymbol{z}') + \Pi_\beta^l(\boldsymbol{z}, \boldsymbol{z}')\Sigma_\beta^l(\boldsymbol{z}, \boldsymbol{z}')\right. \tag{9}$$

$$\left. + \Pi_{\alpha,\beta}^l(\boldsymbol{z}, \boldsymbol{z}')\Sigma_{\alpha,\beta}^l(\boldsymbol{z}, \boldsymbol{z}') + \Pi_{\beta,\alpha}^l(\boldsymbol{z}, \boldsymbol{z}')\Sigma_{\beta,\alpha}^l(\boldsymbol{z}, \boldsymbol{z}')\right) + \Sigma_\alpha^{L+1}(\boldsymbol{z}, \boldsymbol{z}') \tag{10}$$

*where the covariance functions $\Sigma_\alpha^l, \Sigma_\beta^l, \Pi_\alpha^l, \Pi_\beta^l$ are defined in Eq. 4-7.*

The result is proved in the Appendix C, where the detailed recursions of intermediate kernels for the NTK calculation are also presented.

Note that the NTK formula of a complex MLP looks very different from the NTK of a real MLP given by Jacot et al. [2018] due to the existence of interaction between real and imaginary parts, which is caused by joint weight sharing in complex matrix multiplication. However, if we go deeper, does there exist situations that the NTK of a complex MLP will reduce to that of a real MLP?

## 4.2 Asymptotic equivalence of training dynamics

In this subsection, we provide our main results. Surprisingly, we show that, for commonly used complex activation functions, complex networks trained by real-valued backpropagation have the same inductive bias as real networks during training in the infinite-width limit.

We first define asymptotic equivalence between neural networks, which represents a perspective to investigate when will a complex network have different inductive bias from a real network during training. It also helps if we change the network structure or backpropagation algorithms.

**Definition 6** *Two neural networks trained by gradient descent are asymptotic equivalent , if as all widths go to infinity, their neural tangent kernels $\widehat{\Theta}$ converge to the same deterministic limit $\mathring{\Theta}$ at initialization and have the same optimization trajectory during training.*

The following theorem is our main result: if we train complex networks with real-valued back propagation, under very common conditions, the complex MLPs are asymptotic equivalent with real MLPs, thus they have the same training dynamics.

**Theorem 7** *Consider a complex MLP in Eq. (3) and an ordinary real MLP with $L$ hidden layers trained by real-valued backpropagation. Suppose $\sigma_A = \sigma_B$ at initialization and the activation functions satisfy Assumption 2. As the widths go to infinity, they are asymptotic equivalent if the activation functions satisfy one of the following conditions*

> ***Condition 1** All split activation functions $\phi$ satisfying $\phi(\alpha, \beta) = \phi_R(\alpha) + \phi_R(\beta)i$, where $\phi_R$ is a real-valued activation function;*

> ***Condition 2** A subset of holomorphic activation functions $\phi$ satisfying $\phi_2(\alpha, \beta) = \phi_1(\beta, -\alpha)$ and $\frac{\partial \phi_1(\alpha, \beta)}{\partial \beta} = \frac{\partial \phi_2(\alpha, \beta)}{\partial \alpha} = 0$,*

*where the general complex activation function is denoted as $\phi(\alpha, \beta) = \phi_1(\alpha, \beta) + \phi_2(\alpha, \beta)i$ with input pre-activations $\alpha + \beta i$.*

In the Appendix D the result is proved and we also give the sufficient and necessary conditions. For simplicity, here we only show the most informative conditions. The key idea of the proof is transforming asymptotic equivalence into four complex conditions and find the common solutions based on Rules.F.13 in Appendix F.

**Discussion about the Condition 1.** Note that most commonly used complex activation functions satisfy the Condition 1:

$$\phi(z) = \phi_R(\Re(z)) + \phi_R(\Im(z))i$$

like complex sigmoid function [Benvenuto and Piazza, 1992, Nitta, 1997], complex hyperbolic tangent function [Hirose and Yoshida, 2012], etc. It is also worth mentioning that the recently proposed ReLU-based complex activation function $\mathbb{C}$ReLU also satisfies the Condition 1, which has achieved the best performance in feed-forward complex networks in image processing tasks [Trabelsi et al., 2018, Tan et al., 2020] among all ReLU-based complex activation functions.

**Remark 8** *Because of Liouville's theorem, the only complex-valued functions that are bounded and analytic everywhere are constants. Thus in practice, one must choose between boundedness and analyticity for a complex activation function. Before the popularity of ReLU, almost all activation functions in the real case were bounded. Consequently, previous works about complex networks always preferred non-analytic functions to preserve boundedness. Most commonly they applied split activation functions separately to the real and imaginary parts, as investigated in Bassey et al. [2021] and Scardapane et al. [2018]. So the condition contains most complex networks in practice.*

As a result, the theorem demonstrates that for complex networks with all these common complex activations, if they are trained by real-valued BP, then these complex networks reduce to real networks as widths grow, despite the joint interaction weight sharing caused by complex matrix multiplication structure. Consequently, real-valued backpropagation totally eliminates the characteristics of complex networks at infinite width. This may guide the selection of training algorithm in practice if people want to take full advantage of complex networks, and encourage people to explore learning algorithms specially designed for complex networks.

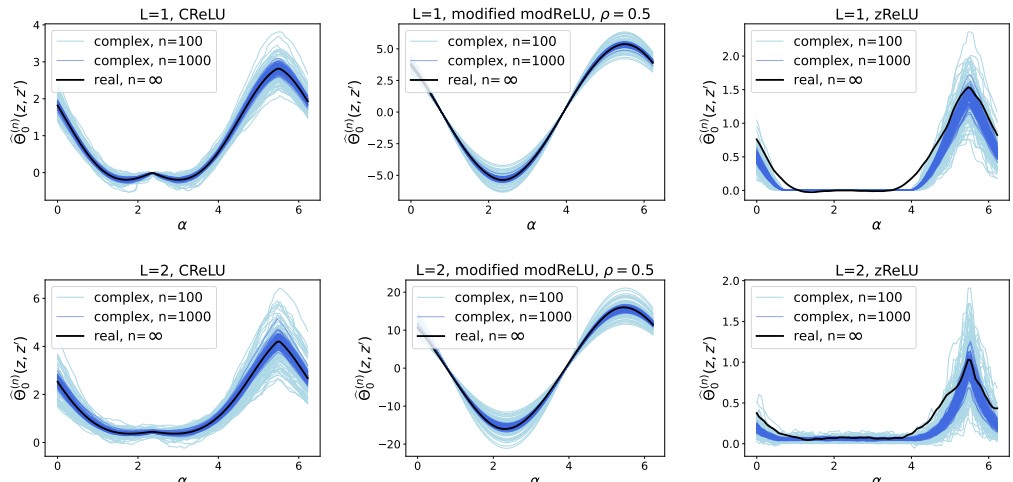

Figure 1: **Convergence of complex NTKs to corresponding real NTKs at initialization.** One input $z = 1 - i$ is fixed and the other input $z' = \cos\alpha + \sin\alpha i$ varies with $\alpha \in [0, 2\pi]$. The black line is the limiting NTKs of real MLPs $\mathring{\Theta}_r(z, z')$ while the light blue and blue ones are empirical NTKs of complex MLPs $\widehat{\Theta}_0^{(n)}(x, x')$ with widths $n = 100$ and 1000. For each width, $\widehat{\Theta}_0^{(n)}(z, z')$ is calculated 100 times randomly, corresponding to 100 lines in the figure.

**Discussion about the Condition 2.** Condition 2 is also non-vacuous since it is a subset of Cauchy-Riemann condition. It includes all the magnitude-based ReLU-type complex activation functions. For example, we can easily obtain a modified modReLU [Arjovsky et al., 2016] and a modified phase-based ReLU satisfying Condition 2 as follows

$$\phi(z) = \begin{cases} z & \text{if } |z| \geq \rho, \\ 0 & \text{otherwise.} \end{cases} \qquad \phi(z) = \begin{cases} z & \text{if } g(\cos\theta_z) \leq \gamma, \\ 0 & \text{otherwise.} \end{cases}$$

where $g(\cos\theta_z)$ can be any function of phase $\cos\theta_z$. Note that, these activation functions are analytic almost everywhere, and according to previous theory, they enjoy better theoretical results like separation results [Zhang et al., 2022] and local minima [Wu et al., 2021]. However, our results indicate that real-valued backpropagation eliminates these advantages of complex networks in the infinite-width limit, which further illustrates the inappropriateness of real-valued backpropagation.

## 5 Empirical study

In this section, we empirically verify the relationship between NTKs of the complex networks and real networks, and investigate the network widths required for the establishment of our results. We consider complex-valued MLPs with one or two hidden layers and we use $\mathbb{C}$ReLU, modified modReLU, $\mathbb{C}$Sigmoid, $\mathbb{C}$tanh and zReLU as the activation functions. Note that all these activation functions satisfy the conditions of our theorem except zReLU. Through the following experiments, we want to check whether the empirical complex NTKs $\widehat{\Theta}_t^{(n)}$ converge to the corresponding real NTKs $\mathring{\Theta}_r$ with different activation functions as the widths $n$ grow.

For real networks, the input is the concatenated vector of the real and imaginary parts of the complex-valued input. Corresponded to the complex-valued fully-connected layer, we use the commonly used real-valued fully-connected layer without complex matrix multiplication structure. To implement the corresponding activation functions, complex-valued activation functions are transformed to real-valued ones in the following way: we divide the pre-activation vector into two half, treat the first half as real parts and the second as imaginary parts, as the input of $\phi(\alpha, \beta)$. Then we concatenate the real and imaginary parts after activation. For split activation functions like $\mathbb{C}$ReLU, it can just correspond to real ReLU activation. In NTK initialization, the standard deviations are set as 1 for complex networks and scaled to $\sqrt{2}$ for real networks. All empirical NTKs of complex networks are calculated based on the Neural Tangents library [Novak et al., 2019].

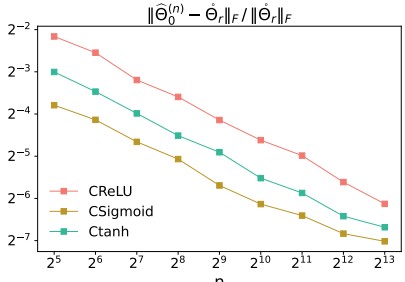 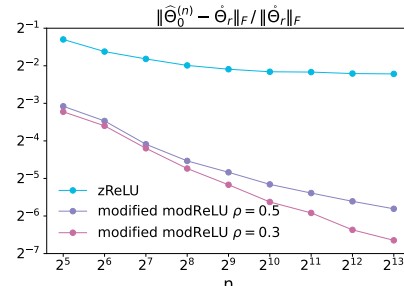

Figure 2: **Convergence of complex NTKs to corresponding real NTKs under various activation functions with a larger range of widths.** The Y-axis is the difference between empirical complex NTKs and the corresponding real NTKs in terms of the relative Frobenius norm. At each point the relative Frobenius norm is calculated 20 times and the mean value is shown. **Left:** three kinds of split activation functions satisfying our conditions. **Right:** other complex-valued activation functions where modified modReLU satisfies conditions and zReLU does not.

**Verifying asymptotic equivalence at initialization.** The first experiment shows the distribution of empirical NTKs of complex MLPs $\widehat{\Theta}_t^{(n)}(z, z')$ and analytic NTKs of real MLPs $\mathring{\Theta}_r(z, z')$ with different $z'$ at initialization on a synthetic dataset. We define $z = 1 - i$ and $z' = \cos\alpha + \sin\alpha i$ for $\alpha \in [0, 2\pi]$. Then we can view $\widehat{\Theta}_t^{(n)}(z, z')$ and $\mathring{\Theta}_r(z, z')$ as functions of $\alpha$. For the complex networks, we calculate empirical NTKs for hidden layer widths $n = 100, 1000$ at initialization and hidden layer number $l = 1, 2$. In each case, we calculate $\widehat{\Theta}_t^{(n)}(z, z')$ 100 times with different random NTK initialization. We compare these empirical complex NTKs with corresponding real NTKs. In the case of $\mathbb{C}$ReLU, we calculate $\mathring{\Theta}_r$ by the analytic form solution of NTK [Cho and Saul, 2009]; in the case of zReLU and modified modReLU, it's hard to get the closed form solution, so we use the average of a large mount of wide empirical NTKs to approximate $\mathring{\Theta}_r$. The results are shown in Figure 1. In the figure we observe that for $\mathbb{C}$ReLU and modified modReLU, which satisfy our conditions, their NTKs $\widehat{\Theta}_0^{(n)}$ concentrate to the NTKs of real MLPs $\mathring{\Theta}_r$ perfectly, and when $n = 1000$, the convergence is more concentrated and the complex NTKs almost equal to real NTKs; for zReLU which does not satisfy the conditions, there's a gap between complex and real NTKs. Therefore, the results verify our theorem perfectly and demonstrate our results are non-vacuous.

**Verifying asymptotic equivalence with more activation functions as widths grow much larger.** In the first experiment, although for zReLU there is a gap between complex and real networks, the tendency is still similar. For the second experiment, we do the similar experiment on the same synthetic dataset, to see what will happen when $n$ becomes much larger, so that it can be more convincing to verify whether it converges. Besides, we consider more different activation functions which satisfy our conditions including $\mathbb{C}$ReLU, $\mathbb{C}$Sigmoid, $\mathbb{C}$tanh and modified modReLU with different hyperparameters $\rho$. We calculate relative Frobenius norm $\|\widehat{\Theta}_0^{(n)}(X, X) - \mathring{\Theta}_r(X, X)\|_F / \|\mathring{\Theta}_r(X, X)\|_F$ on set $X$ with widths $n$ ranging from $2^5$ to $2^{13}$, which measures the difference between complex NTKs $\widehat{\Theta}_0^{(n)}$ and real NTKs $\mathring{\Theta}_r$. Figure 2 shows the result. For all those split activation functions($\mathbb{C}$ReLU, $\mathbb{C}$Sigmoid and $\mathbb{C}$tanh), the tendency of convergence remains unchanged even when the width $n$ goes to quite a large number $2^{13}$. The two curves of modified modReLU act similarly with that of split activation functions; However, the curve of zReLU does not converge at all at initialization.

**Verifying asymptotic equivalence during training.** The third experiment investigates the convergence of difference between complex NTKs $\widehat{\Theta}_t^{(n)}$ and real NTKs $\mathring{\Theta}_r$ during training as the widths go to infinity on MNIST [LeCun et al., 1998]. We randomly choose a subset of MNIST as training set $\mathcal{D} = (X, Y)$ ($|\mathcal{D}| = 128$), and treat the first half of features as real parts, the second half as imaginary parts. Then we calculate relative Frobenius norm between empirical NTKs of complex networks at time $t$ and real NTKs at initialization with widths $n$ ranging from $2^5$ to $2^{10}$ at initialization ($t = 0$) and during training ($t = 1000$). Due to the memory limitations, we cannot try larger $n$. The learning rate $\eta$ is 0.5 for $l = 1$ and 0.2 for $l = 2$. The results are shown in Figure 3. We can see that in all these cases, the relative Frobenius norm decreases as $n$ goes up, regardless of the training steps and

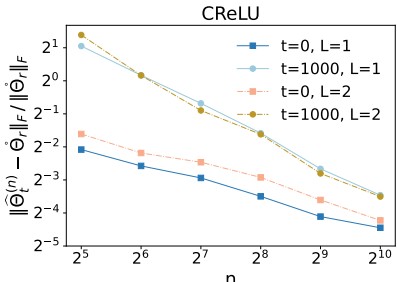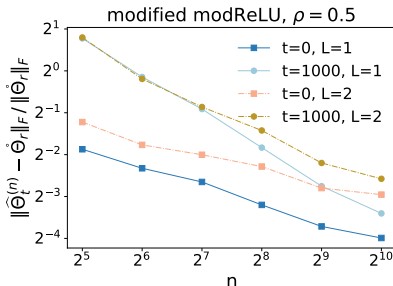

Figure 3: **Convergence of complex NTKs to corresponding real NTKs as widths grow during training.** The Y-axis is the difference between empirical complex NTKs and the corresponding real NTKs in terms of the relative Frobenius norm. At each point the relative Frobenius norm is calculated 20 times and the mean value is shown. The solid line indicates one hidden layer ($L = 1$) while the dashdot line indicates two hidden layers ($L = 2$). The square marker indicates $t = 0$ (at initialization) while the circle marker indicates $t = 1000$ (during training).

hidden layer numbers. Therefore, our theoretical results hold during training. More experiments verifying asymptotic equivalence can be found in Appendix E.

**Empirical results.** Overall, in the case of complex activation functions satisfying our theorems, such as $\mathbb{C}$ReLU and modified modReLU, complex NTKs $\widehat{\Theta}_t^{(n)}$ converges to real NTKs $\mathring{\Theta}_r$ quite well even the widths are about 1000; in the case of complex activation function that does satisfy our theorems like zReLU, the convergence from the complex NTK $\widehat{\Theta}_0^{(n)}$ to the real NTK $\mathring{\Theta}_r$ does not occur. This validates our theory perfectly and demonstrates that our conditions are non-vacuous.

# 6 Related work

Long before the popularity of deep learning, there have been many investigations on complex-valued neural networks [Hirose, 1992, Benvenuto and Piazza, 1992, Nitta, 1997]. However, It is always challenging to train complex networks due to some non-intuitive analytical properties, of which the most notable reason is that almost all cost functions are real-valued and thus non-holomorphic. In order to perform backpropagation for complex networks, the conventional approach to overcome the limitation is to use separate derivatives with respect to the real-imaginary parts of a non-analytic function [Nitta, 2004], or split amplitude-phase parts [Hirose, 1992]. Hirose and Yoshida [2012] has shown that split backpropagation for amplitude-phase parts could achieve better generalization than real networks on signal processing tasks. Recently Wirtinger calculus [Wirtinger, 1927] has received increasing attention [Adali et al., 2011, Bassey et al., 2021], which presents an elegant formulation to derive complex-valued gradient, Jacobian, and Hessian. The real-valued backpropagation [Nitta, 1997] is most widely used, and it becomes essential for training deep complex networks due to the convenience of utilizing the real-valued deep learning library [Arjovsky et al., 2016, Trabelsi et al., 2018, Tan et al., 2020] and achieves state-of-the-art performance. Theoretically, there have been important advances for complex networks regarding the universal approximation property [Voigtlaender, 2020], local minima [Nitta, 2013, Wu et al., 2021], separation results [Zhang et al., 2022]. However, to our best knowledge, there is still no theoretical analysis of the training dynamics of complex networks and the equivalence after training between complex and real networks.

# 7 Conclusion

In this paper, we propose a way to compare the training dynamics between complex and real networks based on their neural tangent kernels (NTKs). Surprisingly, we find that the commonly used real-valued backpropagation reduces the training dynamics of complex-valued MLPs to that of ordinary real MLPs as the widths tend to infinity, thus eliminating the characteristics of complex-valued neural networks. Empirical study verifies that our results are practical for commonly used complex activation functions. The results encourage the design of new training algorithms for complex networks in

the future. Besides, the proposed asymptotic equivalence of training dynamics between networks provides a new perspective for theoretical analysis of neural networks, which may offer a possibility to divide various neural network architectures into equivalence classes.

## Acknowledgment

This research was supported by NSFC (62176117, 61921006) and Collaborative Innovation Center of Novel Software Technology and Industrialization, and the Program B for Outstanding PhD candidate of Nanjing University. The authors would like to thank Jin-Hui Wu and Peng Zhao for helpful discussions. We are also grateful for the anonymous reviewers for their valuable comments.

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
