# A Complex Backpropagation

## A.1 Holomorphic Functions

Consider first a complex-valued function $f(z) = u(x, y) + v(x, y)i$ where $z = x + yi$. The classical definition of *complex differentiability* requires that the derivatives defined as the limit

$$f'(z_0) = \lim_{\Delta z \to 0} \frac{f(z_0 + \Delta z) - f(z_0)}{\Delta z}$$

are independent of the direction in which $\Delta z$ approaches 0 in the complex plane. In order to be complex differentiable, $f(z)$ should satisfy

$$\frac{\partial u}{\partial x} = \frac{\partial v}{\partial y} \text{ and } \frac{\partial u}{\partial y} = -\frac{\partial v}{\partial x}.$$

They are called the Cauchy-Riemann equations, which give a necessary condition for complex differentiability. If the partial derivatives of $u(x, y)$ and $v(x, y)$ are continuous, then the Cauchy-Riemann equations become a sufficient condition as well. A function that is complex differentiable on its entire domain is called *holomorphic* or *analytic*.

## A.2 Real-valued backpropagation for deep complex networks

Obviously, for complex neural networks with real-valued output, the network function $f_\theta(z)$ is not analytic because $v_\theta(x, y) \equiv 0$ and thus the Cauchy-Riemann conditions do not hold. Even for complex neural networks with complex-valued output, in general the Cauchy-Riemann conditions do not hold either because the loss function is real-valued. In order to perform backpropagation in a complex neural network, the conventional approach to overcome this limitation is to use the separate derivatives with respect to the real and imaginary parts of a non-analytic function [Nitta, 1997, Arjovsky et al., 2016, Trabelsi et al., 2018, Zhang and Zhou, 2021]. Thus we only require that the output functions of each layer have continuous partial derivatives with respect to the real and imaginary parts. Such functions are called real-differentiable. For this purpose, it is needed that all the activation functions in the complex networks are real-differentiable.

If $f_\theta(z)$ is a complex network function and $v$ is a complex weight with $v = v_1 + v_2 i$ where $v_1, v_2 \in \mathbb{R}$, then

$$\nabla_v f(z) = \frac{\partial f}{\partial v} = \frac{\partial f}{\partial v_1} + \frac{\partial f}{\partial v_2}i = \Re(\nabla_v f(z)) + \Im(\nabla_v f(z))i.$$

If we have another complex weight $w = w_1 + w_2 i$ where $w_1, w_2 \in \mathbb{R}$ and $v$ could be expressed in terms of $w$, then we have the generalized complex chain rule as follows

$$\begin{aligned}
\nabla_w f(z) = \frac{\partial f}{\partial w} &= \frac{\partial f}{\partial w_1} + \frac{\partial f}{\partial w_2}i = \frac{\partial f}{\partial v}\frac{\partial v}{\partial w_1} + \frac{\partial f}{\partial v}\frac{\partial v}{\partial w_2}i \\
&= \frac{\partial f}{\partial v_1}\frac{\partial v_1}{\partial w_1} + \frac{\partial f}{\partial v_2}\frac{\partial v_2}{\partial w_1} + \left(\frac{\partial f}{\partial v_1}\frac{\partial v_1}{\partial w_2} + \frac{\partial f}{\partial v_2}\frac{\partial v_2}{\partial w_2}\right)i \\
&= \frac{\partial f}{\partial v_1}\left(\frac{\partial v_1}{\partial w_1} + \frac{\partial v_1}{\partial w_2}i\right) + \frac{\partial f}{\partial v_2}\left(\frac{\partial v_2}{\partial w_1} + \frac{\partial v_2}{\partial w_2}i\right) \\
&= \Re(\nabla_w f(z))\left(\frac{\partial v_1}{\partial w_1} + \frac{\partial v_1}{\partial w_2}i\right) + \Im(\nabla_w f(z))\left(\frac{\partial v_2}{\partial w_1} + \frac{\partial v_2}{\partial w_2}i\right)
\end{aligned}$$

# B Proof of the Theorem 3 and the Corollary 4: Complex Tensor Program

For real-valued neural networks that can be expressed by tensor program NETSOR$^\top$ (See Appendix F for details), we have the following theorem from Yang [2020], Corollary 7.3:

**Theorem B.9** *Let $f_{\theta_R}$ be a real-valued (possibly recurrent) neural network with scalar output, which can be expressed by NETSOR$^\top$ and satisfies Condition F.12. If its nonlinearities have polynomially bounded weak derivatives, then its empirical NTK $\widehat{\Theta}$ at initialization converges almost surely, over any finite set of inputs, to a deterministic kernel $\Theta$ as its widths go to infinity and each elements of its factored weights $W$ are randomly initialized as zero-mean Gaussian variables.*

The key idea of the proof is to reduce the complex tensor operations and backward propagation to real-valued tensor program NETSOR$^\top$. First we can easily show that for complex activation functions under Assumption 2, the corresponding real-valued functions are also polynomially-bounded.

**Proposition B.10** *Consider a complex activation function $\phi : \mathbb{C}^k \to \mathbb{C}$ with the input complex vector $\boldsymbol{z} = [z_1, \ldots, z_k]$ where $z_j = x_j + y_j i$ for $\forall j \in [k]$ and $x_j, y_j \in \mathbb{R}$. When $\phi(\boldsymbol{z})$ are polynomially-bounded in the complex domain, if we treat the complex activation function $\phi(\boldsymbol{z})$ as $\phi = \phi_1 + \phi_2 i$ where $\phi_1, \phi_2$ are two real-valued functions with real-valued input $[\boldsymbol{x}, \boldsymbol{y}] \in \mathbb{R}^{2k}$, then the real-valued functions $\phi_1(\boldsymbol{x}, \boldsymbol{y}), \phi_2(\boldsymbol{x}, \boldsymbol{y})$ are both polynomially-bounded.*

**Proof** The proposition can be easily proved if we treat all complex numbers using real values in terms of their real and imaginary parts. According to that $\phi(z)$ satisfies $|\phi(\boldsymbol{z})| \leq C\|\boldsymbol{z}\|^p + c$ for some $C, p, c > 0$ and $\boldsymbol{z} \in \mathbb{C}^k$, let $\boldsymbol{e} = [\boldsymbol{x}, \boldsymbol{y}]$ be the real-valued concatenating vector, we have

$$|\phi(\boldsymbol{z})| = |\phi_1(\boldsymbol{e}) + \phi_2(\boldsymbol{e})i| = \sqrt{\phi_1(\boldsymbol{e})^2 + \phi_2(\boldsymbol{e})^2} \leq C\|\boldsymbol{z}\|^p + c = C\|\boldsymbol{e}\|^p + c$$

Thus obviously we have $|\phi_1(\boldsymbol{e})|, |\phi_2(\boldsymbol{e})| \leq \sqrt{\phi_1(\boldsymbol{e})^2 + \phi_2(\boldsymbol{e})^2} = C\|\boldsymbol{e}\|^p + c$. ∎

**Initial set of random vectors.** Without loss of generality, considering the feed-forward and backward procedure of a complex network, we have initial set of vectors $\{\boldsymbol{\alpha}_1 = \mathbf{A}_1 \boldsymbol{x} - \mathbf{B}_1 \boldsymbol{y}\} \cup \{\boldsymbol{\beta}_1 = \mathbf{A}_1 \boldsymbol{y} + \mathbf{B}_1 \boldsymbol{x}\} \cup \{\boldsymbol{\delta}_s^L(\boldsymbol{z})\} \cup \{\boldsymbol{\delta}_r^L(\boldsymbol{z})\}$ where the gradient vectors $\boldsymbol{\delta}_s^L(\boldsymbol{z})$ and $\boldsymbol{\delta}_r^L(\boldsymbol{z})$ are defined as $\boldsymbol{\delta}_s^L(\boldsymbol{z}) = \sqrt{n}\nabla_{\boldsymbol{s}_L} f(\boldsymbol{z})$ and $\boldsymbol{\delta}_r^L(\boldsymbol{z}) = \sqrt{n}\nabla_{\boldsymbol{r}_L} f(\boldsymbol{z})$ respectively like gradient vectors $\boldsymbol{\delta}_\alpha^l(\boldsymbol{z})$ and $\boldsymbol{\delta}_\beta^l(\boldsymbol{z})$. Since complex NTK parametrization is applied for the weight matrices, it is easy to show that $Z^{\boldsymbol{\alpha}_1}$ and $Z^{\boldsymbol{\beta}_1}$ are distributed as multivariate gaussian over any input instances as follows

$$\left\{ Z^{\boldsymbol{\alpha}_1}, Z^{\boldsymbol{\alpha}_1'} \right\} = \left\{ Z^{\mathbf{A}_1 \boldsymbol{x} - \mathbf{B}_1 \boldsymbol{y}}, Z^{\mathbf{A}_1 \boldsymbol{x}' - \mathbf{B}_1 \boldsymbol{y}'} \right\}$$
$$\sim \mathcal{N}\left( 0, \frac{\sigma_A^2}{d} \begin{pmatrix} \boldsymbol{x}^\top \boldsymbol{x} & \boldsymbol{x}^\top \boldsymbol{x}' \\ \boldsymbol{x}^\top \boldsymbol{x}' & \boldsymbol{x}'^\top \boldsymbol{x}' \end{pmatrix} + \frac{\sigma_B^2}{d} \begin{pmatrix} \boldsymbol{y}^\top \boldsymbol{y} & \boldsymbol{y}^\top \boldsymbol{y}' \\ \boldsymbol{y}^\top \boldsymbol{y}' & \boldsymbol{y}'^\top \boldsymbol{y}' \end{pmatrix} \right).$$
$$\left\{ Z^{\boldsymbol{\beta}_1}, Z^{\boldsymbol{\beta}_1'} \right\} = \left\{ Z^{\mathbf{A}_1 \boldsymbol{y} + \mathbf{B}_1 \boldsymbol{x}}, Z^{\mathbf{A}_1 \boldsymbol{y}' + \mathbf{B}_1 \boldsymbol{x}'} \right\}$$
$$\sim \mathcal{N}\left( 0, \frac{\sigma_B^2}{d} \begin{pmatrix} \boldsymbol{x}^\top \boldsymbol{x} & \boldsymbol{x}^\top \boldsymbol{x}' \\ \boldsymbol{x}^\top \boldsymbol{x}' & \boldsymbol{x}'^\top \boldsymbol{x}' \end{pmatrix} + \frac{\sigma_A^2}{d} \begin{pmatrix} \boldsymbol{y}^\top \boldsymbol{y} & \boldsymbol{y}^\top \boldsymbol{y}' \\ \boldsymbol{y}^\top \boldsymbol{y}' & \boldsymbol{y}'^\top \boldsymbol{y}' \end{pmatrix} \right).$$

where $Z^{\boldsymbol{\alpha}_1}$ and $Z^{\boldsymbol{\beta}_1}$ are random variables corresponding to the iid coordinate of $\boldsymbol{\alpha}_1$ and $\boldsymbol{\beta}_1$. (See details in Rules F.13). In addition, each coordinate of the backward initial vectors $\boldsymbol{\delta}_s^L(\boldsymbol{z}) = \sqrt{n}\nabla_{\boldsymbol{s}_L} f(\boldsymbol{z}) = \sigma_A A_{L+1}$ and $\boldsymbol{\delta}_r^L(\boldsymbol{z}) = \sqrt{n}\nabla_{\boldsymbol{r}_L} f(\boldsymbol{z}) = -\sigma_B B_{L+1}$ are also both gaussian distributed. Therefore the assumptions of Theorem F.14 are satisfied.

**Forward and backward procedure of complex tensor programs.** For a complex-valued neural network $f_\theta(\boldsymbol{z})$, we can always decompose all the complex operations into two-dimensional real-valued operations, denoted as $f_{[\theta_R, \theta_I]}([\boldsymbol{x}, \boldsymbol{y}])$, where $\theta_R, \theta_I \in \mathbb{R}^p$ are the real and imaginary parts of all complex parameters respectively. For the forward procedure of complex tensor programs, given $\mathbf{W} = \mathbf{A} + \mathbf{B}i$ and $\boldsymbol{z} = \boldsymbol{x} + \boldsymbol{y}i$, the complex tensor operation **ComMatMul** $\mathbf{W}\boldsymbol{z} = (\mathbf{A}\boldsymbol{x} - \mathbf{B}\boldsymbol{y}) + (\mathbf{A}\boldsymbol{y} + \mathbf{B}\boldsymbol{x})i$ can be decomposed into four real-valued tensor operations **MatMul** in NETSOR$^\top$ in Definition F.11. And based on the Proposition B.10, a complex activation function can be seen as two real-valued functions with input $(\boldsymbol{x}, \boldsymbol{y})$, thus **ComNonLin** also can be rewritten with real-valued tensor operations **NonLin** in NETSOR$^\top$.

For the backward procedure, without loss of generality, we have the following for the backpropagation of **ComMatMul** and **ComNonLin**

**Nonlin:**
$$\boldsymbol{\delta}_\alpha^l(\boldsymbol{z}) = \boldsymbol{\delta}_s^l(\boldsymbol{z}) \odot \partial\phi_1(\boldsymbol{\alpha}_l + \boldsymbol{\beta}_l i)/\partial\alpha + \boldsymbol{\delta}_r^l(\boldsymbol{z}) \odot \partial\phi_2(\boldsymbol{\alpha}_l + \boldsymbol{\beta}_l i)/\partial\alpha;$$
$$\boldsymbol{\delta}_\beta^l(\boldsymbol{z}) = \boldsymbol{\delta}_s^l(\boldsymbol{z}) \odot \partial\phi_1(\boldsymbol{\alpha}_l + \boldsymbol{\beta}_l i)/\partial\beta + \boldsymbol{\delta}_r^l(\boldsymbol{z}) \odot \partial\phi_2(\boldsymbol{\alpha}_l + \boldsymbol{\beta}_l i)/\partial\beta;$$

**MatMul:**
$$\boldsymbol{\delta}_s^{l-1}(\boldsymbol{z}) = \mathbf{A}_l^\top \boldsymbol{\delta}_\alpha^l(\boldsymbol{z}) + \mathbf{B}_l^\top \boldsymbol{\delta}_\beta^l(\boldsymbol{z});$$
$$\boldsymbol{\delta}_r^{l-1}(\boldsymbol{z}) = -\mathbf{B}_l^\top \boldsymbol{\delta}_\alpha^l(\boldsymbol{z}) + \mathbf{A}_l^\top \boldsymbol{\delta}_\beta^l(\boldsymbol{z}).$$

Therefore, the backward procedure of complex neural networks can also be expressed by NETSOR$^\top$.

Denote the real function corresponding to the complex network as $f_{\theta_r}(\boldsymbol{\tau}) : \mathbb{R}^{2d} \to \mathbb{R}^{d_{out}}$ where the input $\boldsymbol{\tau} = [\Re\{\boldsymbol{z}\}, \Im\{\boldsymbol{z}\}]$ is a real-valued concatenating vector and all weights are decomposed to real-valued matrices. According to the Theorem B.9, for this real-valued network whose forward and backward procedure could be represented by NETSOR$^\top$, its empirical NTK $\widehat{\Theta}_r : \mathbb{R}^{2d} \times \mathbb{R}^{2d} \to \mathbb{R}^{d_{out}}$ defined as

$$\widehat{\Theta}_r(\boldsymbol{\tau}, \boldsymbol{\tau}') = \langle \nabla_{\theta_r} f_{\theta_r}(\boldsymbol{\tau}), \nabla_{\theta_r} f_{\theta_r}(\boldsymbol{\tau}') \rangle$$

converges to a deterministic limiting kernel as widths go to infinity. Therefore, the equivalent complex NTK with corresponding complex input $\widehat{\Theta} : \mathbb{C}^d \times \mathbb{C}^d \to \mathbb{R}^{d_{out}}$ also converges as widths go to infinity. And the limiting NTK could be calculated by Rules F.13. This concludes the proof of Theorem 3.

Based on the Theorem 3 and the main theorem of Yang and Littwin [2021], the proof of Corollary 4 is straight. Due to that complex networks can be decomposed into real operations and represented by NETSOR$^\top$, based on the main theorem in **NTKTRAIN** under the assumptions in Setup D.1 in Yang and Littwin [2021], it can be obtained that the corresponding real function $f_{\theta_r}(\boldsymbol{\tau}) : \mathbb{R}^{2d} \to \mathbb{R}^{d_{out}}$ satisfy Eq. (2)

$$\mathring{f}_{\theta_r, t+1}(\boldsymbol{\tau}) - \mathring{f}_{\theta_r, t}(\boldsymbol{\tau}) = -\mathring{\Theta}_r(\boldsymbol{\tau}, \boldsymbol{\tau}_t) \mathcal{L}'_t\left(\mathring{f}_{\theta_r, t}(\boldsymbol{\tau}_t)\right), \tag{11}$$

where $\Theta_r(\boldsymbol{\tau}, \boldsymbol{\tau}') : \mathbb{R}^{2d} \times \mathbb{R}^{2d} \to \mathbb{R}^{d_{out}}$ is the infinite-width NTK (at initialization) of the corresponding real neural network. Due to the original complex network and complex NTK have the same value as real version, the Corollary 4 is proved.

## C   Proof of Theorem 5: NTK of complex MLPs

First, we consider the case that output dimension $d_{out} = 1$. In this case, the output weight matrix can be written as $\mathbf{W}_{L+1} = \mathbf{A}_{L+1} + \mathbf{B}_{L+1}i = \frac{\sigma_A}{\sqrt{n}} A_{L+1} + \frac{\sigma_B}{\sqrt{n}} B_{L+1}i = \frac{\sigma_A}{\sqrt{n}} \boldsymbol{a} + \frac{\sigma_B}{\sqrt{n}} \boldsymbol{b}i$ for $\boldsymbol{a}, \boldsymbol{b} \in \mathbb{R}^{n \times 1}$ and $f_\theta(\boldsymbol{z}) = \Re\{\mathbf{W}_{L+1} \boldsymbol{h}_L\} = \frac{\sigma_A}{\sqrt{n}} \boldsymbol{a}^\top \boldsymbol{s}_L - \frac{\sigma_B}{\sqrt{n}} \boldsymbol{b}^\top \boldsymbol{r}_L$. Without loss of generality, here we set all the variance of real part and imaginary part as $\sigma_A$ and $\sigma_B$ respectively. We will show that the results of single-dimensional output can be easily extended to multi-dimensional output case.

**Decomposing NTK: The Canonical Decomposition.** Due to that the complex networks are optimized by the generalized BP algorithm, which decomposes the complex weight matrices into real-valued matrices, like Yang [2020], we can first decompose the NTK into contributions from each real-valued parameter's gradient as

$$\begin{aligned}
\widehat{\Theta}(\boldsymbol{z}, \boldsymbol{z}') &= \langle \nabla_\theta f_\theta(\boldsymbol{z}), \nabla_\theta f_\theta(\boldsymbol{z}') \rangle \\
&= \sum_{l=1}^L \langle \nabla_{A_l} f(\boldsymbol{z}), \nabla_{A_l} f(\boldsymbol{z}') \rangle + \sum_{l=1}^L \langle \nabla_{B_l} f(\boldsymbol{z}), \nabla_{B_l} f(\boldsymbol{z}') \rangle \\
&\quad + \langle \nabla_{A_{L+1}} f(\boldsymbol{z}), \nabla_{A_{L+1}} f(\boldsymbol{z}') \rangle + \langle \nabla_{B_{L+1}} f(\boldsymbol{z}), \nabla_{B_{L+1}} f(\boldsymbol{z}') \rangle
\end{aligned} \tag{12}$$

Considering that for $l \in [1, L]$, we have

$$\boldsymbol{s}_l + \boldsymbol{r}_l i = \phi((\mathbf{A}_l + \mathbf{B}_l i)(\boldsymbol{s}_{l-1} + \boldsymbol{r}_{l-1} i)) = \phi((\mathbf{A}_l \boldsymbol{s}_{l-1} - \mathbf{B}_l \boldsymbol{r}_{l-1}) + (\mathbf{A}_l \boldsymbol{r}_{l-1} + \mathbf{B}_l \boldsymbol{s}_{l-1} i)) \tag{13}$$

Based on gradient vectors $\boldsymbol{\delta}_\alpha^l(\boldsymbol{z})$ and $\boldsymbol{\delta}_\beta^l(\boldsymbol{z})$, when $l \in [2, L]$, the first term of Eq. (12) can be written as

$$\begin{aligned}
\langle \nabla_{A_l} f(\boldsymbol{z}), \nabla_{A_l} f(\boldsymbol{z}') \rangle &= \frac{\sigma_A^2}{n} \langle \nabla_{\mathbf{A}_l} f(\boldsymbol{z}), \nabla_{\mathbf{A}_l} f(\boldsymbol{z}') \rangle \\
&= \frac{\sigma_A^2}{n^2} \left\langle \boldsymbol{\delta}_\alpha^l(\boldsymbol{z}) \boldsymbol{s}_{l-1}^\top + \boldsymbol{\delta}_\beta^l(\boldsymbol{z}) \boldsymbol{r}_{l-1}^\top, \boldsymbol{\delta}_\alpha^l(\boldsymbol{z}') {\boldsymbol{s}'_{l-1}}^\top + \boldsymbol{\delta}_\beta^l(\boldsymbol{z}') {\boldsymbol{r}'_{l-1}}^\top \right\rangle \\
&= \sigma_A^2 \frac{\boldsymbol{\delta}_\alpha^l(\boldsymbol{z})^\top \boldsymbol{\delta}_\alpha^l(\boldsymbol{z}')}{n} \frac{\boldsymbol{s}_{l-1}^\top \boldsymbol{s}'_{l-1}}{n} + \sigma_A^2 \frac{\boldsymbol{\delta}_\beta^l(\boldsymbol{z})^\top \boldsymbol{\delta}_\beta^l(\boldsymbol{z}')}{n} \frac{\boldsymbol{r}_{l-1}^\top \boldsymbol{r}'_{l-1}}{n} \\
&\quad + \sigma_A^2 \frac{\boldsymbol{\delta}_\alpha^l(\boldsymbol{z})^\top \boldsymbol{\delta}_\beta^l(\boldsymbol{z}')}{n} \frac{\boldsymbol{s}_{l-1}^\top \boldsymbol{r}'_{l-1}}{n} + \sigma_A^2 \frac{\boldsymbol{\delta}_\beta^l(\boldsymbol{z})^\top \boldsymbol{\delta}_\alpha^l(\boldsymbol{z}')}{n} \frac{\boldsymbol{r}_{l-1}^\top \boldsymbol{s}'_{l-1}}{n}
\end{aligned} \tag{14}$$

Note that for simplicity, here we abbreviate $s_l(z')$ and $r_l(z')$ as $s_l'$ and $r_l'$ respectively. Then for the second term, when $l \in [2, L]$, we have decomposition as follows

$$
\begin{aligned}
\langle \nabla_{B_l} f(z), \nabla_{B_l} f(z') \rangle &= \frac{\sigma_B^2}{n} \langle \nabla_{\mathbf{B}_l} f(z), \nabla_{\mathbf{B}_l} f(z') \rangle \\
&= \frac{\sigma_B^2}{n^2} \left\langle -\delta_\alpha^l(z) r_{l-1}^\top + \delta_\beta^l(z) s_{l-1}^\top, -\delta_\alpha^l(z') r_{l-1}'^\top + \delta_\beta^l(z') s_{l-1}'^\top \right\rangle \\
&= \sigma_B^2 \frac{\delta_\alpha^l(z)^\top \delta_\alpha^l(z')}{n} \frac{r_{l-1}^\top r_{l-1}'}{n} + \sigma_B^2 \frac{\delta_\beta^l(z)^\top \delta_\beta^l(z')}{n} \frac{s_{l-1}^\top s_{l-1}'}{n} \\
&\quad - \sigma_B^2 \frac{\delta_\alpha^l(z)^\top \delta_\beta^l(z')}{n} \frac{r_{l-1}^\top s_{l-1}'}{n} - \sigma_B^2 \frac{\delta_\beta^l(z)^\top \delta_\alpha^l(z')}{n} \frac{s_{l-1}^\top r_{l-1}'}{n} \quad (15)
\end{aligned}
$$

When $l = 1$, i.e., for the input layer, since the weight matrix is $\mathbb{C}^{n \times d}$, thus we have the following decomposition for the first and second term in Eq. (12)

$$
\begin{aligned}
\langle \nabla_{A_1} f(z), \nabla_{A_1} f(z') \rangle &= \sigma_A^2 \frac{\delta_\alpha^1(z)^\top \delta_\alpha^1(z')}{n} \frac{x^\top x'}{d} + \sigma_A^2 \frac{\delta_\beta^1(z)^\top \delta_\beta^1(z')}{n} \frac{y^\top y'}{d} \\
&\quad + \sigma_A^2 \frac{\delta_\alpha^1(z)^\top \delta_\beta^1(z')}{n} \frac{x^\top y'}{d} + \sigma_A^2 \frac{\delta_\beta^1(z)^\top \delta_\alpha^1(z')}{n} \frac{y^\top x'}{d} \quad (16)
\end{aligned}
$$

$$
\begin{aligned}
\langle \nabla_{B_1} f(z), \nabla_{B_1} f(z') \rangle &= \sigma_B^2 \frac{\delta_\alpha^1(z)^\top \delta_\alpha^1(z')}{n} \frac{y^\top y'}{d} + \sigma_B^2 \frac{\delta_\beta^1(z)^\top \delta_\beta^1(z')}{n} \frac{x^\top x'}{d} \\
&\quad - \sigma_B^2 \frac{\delta_\alpha^1(z)^\top \delta_\beta^1(z')}{n} \frac{y^\top x'}{d} - \sigma_B^2 \frac{\delta_\beta^1(z)^\top \delta_\alpha^1(z')}{n} \frac{x^\top y'}{d} \quad (17)
\end{aligned}
$$

Finally, for the term of output layer, we can obtain

$$
\langle \nabla_{A_{L+1}} f(z), \nabla_{A_{L+1}} f(z') \rangle = \frac{\sigma_A^2}{n} \langle \nabla_{\mathbf{A}_{L+1}} f(z), \nabla_{\mathbf{A}_{L+1}} f(z') \rangle = \sigma_A^2 \frac{s_L^\top s_L'}{n} \quad (18)
$$

$$
\langle \nabla_{B_{L+1}} f(z), \nabla_{B_{L+1}} f(z') \rangle = \frac{\sigma_B^2}{n} \langle \nabla_{\mathbf{B}_{L+1}} f(z), \nabla_{\mathbf{B}_{L+1}} f(z') \rangle = \sigma_B^2 \frac{r_L^\top r_L'}{n} \quad (19)
$$

**Initial set of random vectors.** Considering the feed-forward and backward procedure of complex full-connected networks, we have initial set of vectors $\{\alpha_1 = \mathbf{A}_1 x - \mathbf{B}_1 y\} \cup \{\beta_1 = \mathbf{A}_1 y + \mathbf{B}_1 x\} \cup \{\delta_s^L(z)\} \cup \{\delta_r^L(z)\}$ where the gradient vectors $\delta_s^L(z)$ and $\delta_r^L(z)$ are defined as $\delta_s^L(z) = \sqrt{n} \nabla_{s_L} f(z)$ and $\delta_r^L(z) = \sqrt{n} \nabla_{r_L} f(z)$ respectively like gradient vectors $\delta_\alpha^l(z)$ and $\delta_\beta^l(z)$. Since complex NTK initialization is applied for the weight matrices, it is easy to show that $Z^{\alpha_1}$ and $Z^{\beta_1}$ are distributed as multivariate gaussian over any input instances as follows

$$
\begin{aligned}
\left\{ Z^{\alpha_1}, Z^{\alpha_1'} \right\} &= \left\{ Z^{\mathbf{A}_1 x - \mathbf{B}_1 y}, Z^{\mathbf{A}_1 x' - \mathbf{B}_1 y'} \right\} \\
&\sim \mathcal{N} \left( 0, \frac{\sigma_A^2}{d} \begin{pmatrix} x^\top x & x^\top x' \\ x^\top x' & x'^\top x' \end{pmatrix} + \frac{\sigma_B^2}{d} \begin{pmatrix} y^\top y & y^\top y' \\ y^\top y' & y'^\top y' \end{pmatrix} \right).
\end{aligned}
$$

$$
\begin{aligned}
\left\{ Z^{\beta_1}, Z^{\beta_1'} \right\} &= \left\{ Z^{\mathbf{A}_1 y + \mathbf{B}_1 x}, Z^{\mathbf{A}_1 y' + \mathbf{B}_1 x'} \right\} \\
&\sim \mathcal{N} \left( 0, \frac{\sigma_B^2}{d} \begin{pmatrix} x^\top x & x^\top x' \\ x^\top x' & x'^\top x' \end{pmatrix} + \frac{\sigma_A^2}{d} \begin{pmatrix} y^\top y & y^\top y' \\ y^\top y' & y'^\top y' \end{pmatrix} \right).
\end{aligned}
$$

where $Z^{\alpha_1}$ and $Z^{\beta_1}$ are random variables corresponding to the iid coordinate of $\alpha_1$ and $\beta_1$. (See details in Rules F.13). In addition, each coordinate of the backward initial vectors $\delta_s^L(z) = \sqrt{n} \nabla_{s_L} f(z) = \sigma_A A_{L+1}$ and $\delta_r^L(z) = \sqrt{n} \nabla_{r_L} f(z) = -\sigma_B B_{L+1}$ are also both gaussian distributed. Therefore the assumptions of Theorem F.14 are satisfied.

**Formulation of corresponding NETSOR$^\top$ program.** As proved in the Theorem 3, a complex network which can be represented by complex tensor program could also be expressed by the NETSOR$^\top$. The complex full-connected network can be represented as the following NETSOR$^\top$ program, where we denote the corresponding real activation functions with real-valued input $[\alpha_l, \beta_l]$ as $\widetilde{\phi}_1$ and $\widetilde{\phi}_2$.

**Nonlin:**

$$s_l = \phi_1(\alpha_l + \beta_l i) = \phi_1((\alpha_A^l - \alpha_B^l) + (\beta_A^l + \beta_B^l))i) = \widetilde{\phi}_1(\alpha_l, \beta_l);$$

$$r_l = \phi_2(\alpha_l + \beta_l i) = \phi_2((\alpha_A^l - \alpha_B^l) + (\beta_A^l + \beta_B^l))i) = \widetilde{\phi}_2(\alpha_l, \beta_l);$$

**MatMul:**

$$\alpha_A^l = \mathbf{A}_l s_{l-1}; \quad \alpha_B^l = \mathbf{B}_l r_{l-1}; \quad \beta_A^l = \mathbf{A}_l r_{l-1}; \quad \beta_B^l = \mathbf{B}_l s_{l-1};$$

**Nonlin:**

$$\delta_\alpha^l(z) = \delta_s^l(z) \odot \partial \widetilde{\phi}_1(\alpha_l, \beta_l)/\partial \alpha_l + \delta_r^l(z) \odot \partial \widetilde{\phi}_2(\alpha_l, \beta_l)/\partial \alpha_l;$$

$$\delta_\beta^l(z) = \delta_s^l(z) \odot \partial \widetilde{\phi}_1(\alpha_l, \beta_l)/\partial \beta_l + \delta_r^l(z) \odot \partial \widetilde{\phi}_2(\alpha_l, \beta_l)/\partial \beta_l;$$

**MatMul:**

$$\delta_s^{l-1}(z) = \mathbf{A}_l^\top \delta_\alpha^l(z) + \mathbf{B}_l^\top \delta_\beta^l(z);$$

$$\delta_r^{l-1}(z) = -\mathbf{B}_l^\top \delta_\alpha^l(z) + \mathbf{A}_l^\top \delta_\beta^l(z).$$

**Convergence of intermediate kernels.** Based on Proposition B.10, it is known that the activation functions under Assumption 2 satisfies that the corresponding real activation functions $\phi_1$ and $\phi_2$ are polynomially bounded. The corresponding NETSOR$^\top$ program satisfies the Simple GIA Check (See details in Condition F.12), thus by recursively applying the Master Theorem F.14 we can obtain that as the width $n \to \infty$, all the intermediate kernels $\Sigma_\alpha^l, \Sigma_\beta^l, \Pi_\alpha^l, \Pi_\beta^l, \Sigma_{\alpha,\beta}^l, \Sigma_{\beta,\alpha}^l, \Pi_{\alpha,\beta}^l, \Pi_{\beta,\alpha}^l$ converge to the limits defined in Rules F.13. Specifically, for $\forall l \in [L]$, we have the following limits for the intermediate kernels:

$$\lim_{n\to\infty} \Sigma_\alpha^l(z, z') \xrightarrow{\text{a.s.}} \mathbb{E}Z^{\alpha_l} Z^{\alpha'_l} \qquad\qquad \lim_{n\to\infty} \Sigma_{\alpha,\beta}^l(z, z') \xrightarrow{\text{a.s.}} \mathbb{E}Z^{\alpha_l} Z^{\beta'_l}$$

$$\lim_{n\to\infty} \Sigma_\beta^l(z, z') \xrightarrow{\text{a.s.}} \mathbb{E}Z^{\beta_l} Z^{\beta'_l} \qquad\qquad \lim_{n\to\infty} \Sigma_{\beta,\alpha}^l(z, z') \xrightarrow{\text{a.s.}} \mathbb{E}Z^{\beta_l} Z^{\alpha'_l}$$

$$\lim_{n\to\infty} \Pi_\alpha^l(z, z') \xrightarrow{\text{a.s.}} \mathbb{E}Z^{\delta_\alpha^l(z)} Z^{\delta_\alpha^l(z')} \qquad\qquad \lim_{n\to\infty} \Pi_{\alpha,\beta}^l(z, z') \xrightarrow{\text{a.s.}} \mathbb{E}Z^{\delta_\alpha^l(z)} Z^{\delta_\beta^l(z')}$$

$$\lim_{n\to\infty} \Pi_\beta^l(z, z') \xrightarrow{\text{a.s.}} \mathbb{E}Z^{\delta_\beta^l(z)} Z^{\delta_\beta^l(z')} \qquad\qquad \lim_{n\to\infty} \Pi_{\beta,\alpha}^l(z, z') \xrightarrow{\text{a.s.}} \mathbb{E}Z^{\delta_\beta^l(z)} Z^{\delta_\alpha^l(z')}$$

where $Z^{\alpha_l}, Z^{\beta_l}, Z^{\delta_\alpha^l(z)}$ and $Z^{\delta_\beta^l(z)}$ are random variables corresponding to the iid coordinate defined in Rules F.13.

**Intermediate kernels of forward procedure.** The vectors $s_l$, $r_l$, $\alpha_l$ and $\beta_l$ have roughly iid coordinates distributed as $Z^{s_l}$, $Z^{r_l}$, $Z^{\alpha_l}$ and $Z^{\beta_l}$ respectively. As all widths go to infinity, based on the Rules F.13, we can make the following calculations for the forward iteration of kernels.

$$\Sigma_\alpha^l(z, z') = \mathbb{E}Z^{\alpha_l} Z^{\alpha'_l} = \sigma_A^2 \mathbb{E}Z^{s_{l-1}} Z^{s'_{l-1}} + \sigma_B^2 \mathbb{E}Z^{r_{l-1}} Z^{r'_{l-1}} \tag{20}$$

$$\Sigma_\beta^l(z, z') = \mathbb{E}Z^{\beta_l} Z^{\beta'_l} = \sigma_A^2 \mathbb{E}Z^{r_{l-1}} Z^{r'_{l-1}} + \sigma_B^2 \mathbb{E}Z^{s_{l-1}} Z^{s'_{l-1}} \tag{21}$$

$$\Sigma_{\alpha,\beta}^l(z, z') = \mathbb{E}Z^{\alpha_l} Z^{\beta'_l} = \sigma_A^2 \mathbb{E}Z^{s_{l-1}} Z^{r'_{l-1}} - \sigma_B^2 \mathbb{E}Z^{r_{l-1}} Z^{s'_{l-1}} \tag{22}$$

$$\Sigma_{\beta,\alpha}^l(z, z') = \mathbb{E}Z^{\beta_l} Z^{\alpha'_l} = \sigma_A^2 \mathbb{E}Z^{r_{l-1}} Z^{s'_{l-1}} - \sigma_B^2 \mathbb{E}Z^{s_{l-1}} Z^{r'_{l-1}} \tag{23}$$

Then, for $Z^{s_l}$ and $Z^{r_l}$, we can obtain

$$\mathbb{E}Z^{s_l} Z^{s'_l} = \mathbb{E}Z^{\phi_1(\alpha_l + \beta_l i)} Z^{\phi_1(\alpha'_l + \beta'_l i)} = \mathbb{E}Z^{\widetilde{\phi}_1(\alpha_l, \beta_l)} Z^{\widetilde{\phi}_1(\alpha'_l, \beta'_l)}$$

$$= \mathbb{E}\widetilde{\phi}_1(Z^{\alpha_l}, Z^{\beta_l})\widetilde{\phi}_1(Z^{\alpha'_l}, Z^{\beta'_l}) = \mathbb{E}\widetilde{\phi}_1(\xi_l, \zeta_l)\widetilde{\phi}_1(\xi'_l, \zeta'_l) \tag{24}$$

$$\mathbb{E}Z^{r_l} Z^{r'_l} = \mathbb{E}Z^{\phi_2(\alpha_l + \beta_l i)} Z^{\phi_2(\alpha'_l + \beta'_l i)} = \mathbb{E}Z^{\widetilde{\phi}_2(\alpha_l, \beta_l)} Z^{\widetilde{\phi}_2(\alpha'_l, \beta'_l)}$$

$$= \mathbb{E}\widetilde{\phi}_2(Z^{\alpha_l}, Z^{\beta_l})\widetilde{\phi}_2(Z^{\alpha'_l}, Z^{\beta'_l}) = \mathbb{E}\widetilde{\phi}_2(\xi_l, \zeta_l)\widetilde{\phi}_2(\xi'_l, \zeta'_l) \tag{25}$$

Similarly, we have $\mathbb{E}Z^{s_l} Z^{r'_l} = \mathbb{E}\widetilde{\phi}_1(\xi_l, \zeta_l)\widetilde{\phi}_2(\xi'_l, \zeta'_l)$ and $\mathbb{E}Z^{r_l} Z^{s'_l} = \mathbb{E}\widetilde{\phi}_2(\xi_l, \zeta_l)\widetilde{\phi}_1(\xi'_l, \zeta'_l)$ where the random variables $\xi_l, \zeta_l, \xi'_l, \zeta'_l$ satisfy

$$(\xi_l, \zeta_l, \xi'_l, \zeta'_l) \sim \mathcal{N}\left(0, \begin{pmatrix} \Sigma_\alpha^l(z,z) & \Sigma_{\alpha,\beta}^l(z,z) & \Sigma_\alpha^l(z,z') & \Sigma_{\alpha,\beta}^l(z,z') \\ \Sigma_{\beta,\alpha}^l(z,z) & \Sigma_\beta^l(z,z) & \Sigma_{\beta,\alpha}^l(z,z') & \Sigma_\beta^l(z,z') \\ \Sigma_\alpha^l(z',z) & \Sigma_{\alpha,\beta}^l(z',z) & \Sigma_\alpha^l(z',z') & \Sigma_{\alpha,\beta}^l(z',z') \\ \Sigma_{\beta,\alpha}^l(z',z) & \Sigma_\beta^l(z',z) & \Sigma_{\beta,\alpha}^l(z',z') & \Sigma_\beta^l(z',z') \end{pmatrix}\right) \tag{26}$$

Thus the covariance matrix of each two variables among $\xi_l, \zeta_l, \xi_l', \zeta_l'$ can be directly obtained from Eq.(26).

**Intermediate kernels of backward procedure.** Following the derivation of the feed-forward limiting kernels, it is known that the vectors $\boldsymbol{\delta}_s^l(\boldsymbol{z})$, $\boldsymbol{\delta}_r^l(\boldsymbol{z})$, $\boldsymbol{\delta}_\alpha^l(\boldsymbol{z})$ and $\boldsymbol{\delta}_\beta^l(\boldsymbol{z})$ have roughly iid coordinates distributed as $Z^{\boldsymbol{\delta}_s^l(\boldsymbol{z})}$, $Z^{\boldsymbol{\delta}_r^l(\boldsymbol{z})}$, $Z^{\boldsymbol{\delta}_\alpha^l(\boldsymbol{z})}$ and $Z^{\boldsymbol{\delta}_\beta^l(\boldsymbol{z})}$ respectively. As all widths go to infinity, based on the Rules F.13, we can make the following calculations for the backward iteration of kernels.

$$\mathbb{E}Z^{\boldsymbol{\delta}_s^{l-1}(\boldsymbol{z})}Z^{\boldsymbol{\delta}_s^{l-1}(\boldsymbol{z}')} = \sigma_A^2 \mathbb{E}Z^{\boldsymbol{\delta}_\alpha^l(\boldsymbol{z})}Z^{\boldsymbol{\delta}_\alpha^l(\boldsymbol{z}')} + \sigma_B^2 \mathbb{E}Z^{\boldsymbol{\delta}_\beta^l(\boldsymbol{z})}Z^{\boldsymbol{\delta}_\beta^l(\boldsymbol{z}')} \tag{27}$$

$$\mathbb{E}Z^{\boldsymbol{\delta}_r^{l-1}(\boldsymbol{z})}Z^{\boldsymbol{\delta}_r^{l-1}(\boldsymbol{z}')} = \sigma_A^2 \mathbb{E}Z^{\boldsymbol{\delta}_\beta^l(\boldsymbol{z})}Z^{\boldsymbol{\delta}_\beta^l(\boldsymbol{z}')} + \sigma_B^2 \mathbb{E}Z^{\boldsymbol{\delta}_\alpha^l(\boldsymbol{z})}Z^{\boldsymbol{\delta}_\alpha^l(\boldsymbol{z}')} \tag{28}$$

$$\mathbb{E}Z^{\boldsymbol{\delta}_s^{l-1}(\boldsymbol{z})}Z^{\boldsymbol{\delta}_r^{l-1}(\boldsymbol{z}')} = \sigma_A^2 \mathbb{E}Z^{\boldsymbol{\delta}_\alpha^l(\boldsymbol{z})}Z^{\boldsymbol{\delta}_\beta^l(\boldsymbol{z}')} - \sigma_B^2 \mathbb{E}Z^{\boldsymbol{\delta}_\beta^l(\boldsymbol{z})}Z^{\boldsymbol{\delta}_\alpha^l(\boldsymbol{z}')} \tag{29}$$

$$\mathbb{E}Z^{\boldsymbol{\delta}_r^{l-1}(\boldsymbol{z})}Z^{\boldsymbol{\delta}_s^{l-1}(\boldsymbol{z}')} = \sigma_A^2 \mathbb{E}Z^{\boldsymbol{\delta}_\beta^l(\boldsymbol{z})}Z^{\boldsymbol{\delta}_\alpha^l(\boldsymbol{z}')} - \sigma_B^2 \mathbb{E}Z^{\boldsymbol{\delta}_\alpha^l(\boldsymbol{z})}Z^{\boldsymbol{\delta}_\beta^l(\boldsymbol{z}')} \tag{30}$$

Then, for $Z^{\boldsymbol{\delta}_\alpha^l(\boldsymbol{z})}$ and $Z^{\boldsymbol{\delta}_\beta^l(\boldsymbol{z})}$, we can obtain

$$\Pi_\alpha^l(\boldsymbol{z}, \boldsymbol{z}') = \mathbb{E}Z^{\boldsymbol{\delta}_\alpha^l(\boldsymbol{z})}Z^{\boldsymbol{\delta}_\alpha^l(\boldsymbol{z}')}$$
$$= \mathbb{E}Z^{\boldsymbol{\delta}_s^l(\boldsymbol{z})\odot\partial\widetilde{\phi}_1(\boldsymbol{\alpha}_l,\boldsymbol{\beta}_l)/\partial\boldsymbol{\alpha}_l + \boldsymbol{\delta}_r^l(\boldsymbol{z})\odot\partial\widetilde{\phi}_2(\boldsymbol{\alpha}_l,\boldsymbol{\beta}_l)/\partial\boldsymbol{\alpha}_l} Z^{\boldsymbol{\delta}_s^l(\boldsymbol{z}')\odot\partial\widetilde{\phi}_1(\boldsymbol{\alpha}_l',\boldsymbol{\beta}_l')/\partial\boldsymbol{\alpha}_l' + \boldsymbol{\delta}_r^l(\boldsymbol{z}')\odot\partial\widetilde{\phi}_2(\boldsymbol{\alpha}_l',\boldsymbol{\beta}_l')/\partial\boldsymbol{\alpha}_l'}$$
$$= \mathbb{E}Z^{\boldsymbol{\delta}_s^l(\boldsymbol{z})}Z^{\boldsymbol{\delta}_s^l(\boldsymbol{z}')}\mathbb{E}Z^{\partial\widetilde{\phi}_1(\boldsymbol{\alpha}_l,\boldsymbol{\beta}_l)/\partial\boldsymbol{\alpha}_l}Z^{\partial\widetilde{\phi}_1(\boldsymbol{\alpha}_l',\boldsymbol{\beta}_l')/\partial\boldsymbol{\alpha}_l'}$$
$$\quad + \mathbb{E}Z^{\boldsymbol{\delta}_r^l(\boldsymbol{z})}Z^{\boldsymbol{\delta}_r^l(\boldsymbol{z}')}\mathbb{E}Z^{\partial\widetilde{\phi}_2(\boldsymbol{\alpha}_l,\boldsymbol{\beta}_l)/\partial\boldsymbol{\alpha}_l}Z^{\partial\widetilde{\phi}_2(\boldsymbol{\alpha}_l',\boldsymbol{\beta}_l')/\partial\boldsymbol{\alpha}_l'}$$
$$\quad + \mathbb{E}Z^{\boldsymbol{\delta}_s^l(\boldsymbol{z})}Z^{\boldsymbol{\delta}_r^l(\boldsymbol{z}')}\mathbb{E}Z^{\partial\widetilde{\phi}_1(\boldsymbol{\alpha}_l,\boldsymbol{\beta}_l)/\partial\boldsymbol{\alpha}_l}Z^{\partial\widetilde{\phi}_2(\boldsymbol{\alpha}_l',\boldsymbol{\beta}_l')/\partial\boldsymbol{\alpha}_l'}$$
$$\quad + \mathbb{E}Z^{\boldsymbol{\delta}_r^l(\boldsymbol{z})}Z^{\boldsymbol{\delta}_s^l(\boldsymbol{z}')}\mathbb{E}Z^{\partial\widetilde{\phi}_2(\boldsymbol{\alpha}_l,\boldsymbol{\beta}_l)/\partial\boldsymbol{\alpha}_l}Z^{\partial\widetilde{\phi}_1(\boldsymbol{\alpha}_l',\boldsymbol{\beta}_l')/\partial\boldsymbol{\alpha}_l'} \tag{31}$$
$$= \mathbb{E}Z^{\boldsymbol{\delta}_s^l(\boldsymbol{z})}Z^{\boldsymbol{\delta}_s^l(\boldsymbol{z}')}\mathbb{E}\frac{\partial\widetilde{\phi}_1(\xi_l,\zeta_l)}{\partial\xi_l}\frac{\partial\widetilde{\phi}_1(\xi_l',\zeta_l')}{\partial\xi_l'} + \mathbb{E}Z^{\boldsymbol{\delta}_r^l(\boldsymbol{z})}Z^{\boldsymbol{\delta}_r^l(\boldsymbol{z}')}\mathbb{E}\frac{\partial\widetilde{\phi}_2(\xi_l,\zeta_l)}{\partial\xi_l}\frac{\partial\widetilde{\phi}_2(\xi_l',\zeta_l')}{\partial\xi_l'}$$
$$+ \mathbb{E}Z^{\boldsymbol{\delta}_s^l(\boldsymbol{z})}Z^{\boldsymbol{\delta}_r^l(\boldsymbol{z}')}\mathbb{E}\frac{\partial\widetilde{\phi}_1(\xi_l,\zeta_l)}{\partial\xi_l}\frac{\partial\widetilde{\phi}_2(\xi_l',\zeta_l')}{\partial\xi_l'} + \mathbb{E}Z^{\boldsymbol{\delta}_r^l(\boldsymbol{z})}Z^{\boldsymbol{\delta}_s^l(\boldsymbol{z}')}\mathbb{E}\frac{\partial\widetilde{\phi}_2(\xi_l,\zeta_l)}{\partial\xi_l}\frac{\partial\widetilde{\phi}_1(\xi_l',\zeta_l')}{\partial\xi_l'} \tag{32}$$

Similarly, we have

$$\Pi_\beta^l(\boldsymbol{z}, \boldsymbol{z}') = \mathbb{E}Z^{\boldsymbol{\delta}_\beta^l(\boldsymbol{z})}Z^{\boldsymbol{\delta}_\beta^l(\boldsymbol{z}')}$$
$$= \mathbb{E}Z^{\boldsymbol{\delta}_s^l(\boldsymbol{z})\odot\partial\widetilde{\phi}_1(\boldsymbol{\alpha}_l,\boldsymbol{\beta}_l)/\partial\boldsymbol{\beta}_l + \boldsymbol{\delta}_r^l(\boldsymbol{z})\odot\partial\widetilde{\phi}_2(\boldsymbol{\alpha}_l,\boldsymbol{\beta}_l)/\partial\boldsymbol{\beta}_l} Z^{\boldsymbol{\delta}_s^l(\boldsymbol{z}')\odot\partial\widetilde{\phi}_1(\boldsymbol{\alpha}_l',\boldsymbol{\beta}_l')/\partial\boldsymbol{\beta}_l' + \boldsymbol{\delta}_r^l(\boldsymbol{z}')\odot\partial\widetilde{\phi}_2(\boldsymbol{\alpha}_l',\boldsymbol{\beta}_l')/\partial\boldsymbol{\beta}_l'}$$
$$= \mathbb{E}Z^{\boldsymbol{\delta}_s^l(\boldsymbol{z})}Z^{\boldsymbol{\delta}_s^l(\boldsymbol{z}')}\mathbb{E}\frac{\partial\widetilde{\phi}_1(\xi_l,\zeta_l)}{\partial\zeta_l}\frac{\partial\widetilde{\phi}_1(\xi_l',\zeta_l')}{\partial\zeta_l'} + \mathbb{E}Z^{\boldsymbol{\delta}_r^l(\boldsymbol{z})}Z^{\boldsymbol{\delta}_r^l(\boldsymbol{z}')}\mathbb{E}\frac{\partial\widetilde{\phi}_2(\xi_l,\zeta_l)}{\partial\zeta_l}\frac{\partial\widetilde{\phi}_2(\xi_l',\zeta_l')}{\partial\zeta_l'}$$
$$+ \mathbb{E}Z^{\boldsymbol{\delta}_s^l(\boldsymbol{z})}Z^{\boldsymbol{\delta}_r^l(\boldsymbol{z}')}\mathbb{E}\frac{\partial\widetilde{\phi}_1(\xi_l,\zeta_l)}{\partial\zeta_l}\frac{\partial\widetilde{\phi}_2(\xi_l',\zeta_l')}{\partial\zeta_l'} + \mathbb{E}Z^{\boldsymbol{\delta}_r^l(\boldsymbol{z})}Z^{\boldsymbol{\delta}_s^l(\boldsymbol{z}')}\mathbb{E}\frac{\partial\widetilde{\phi}_2(\xi_l,\zeta_l)}{\partial\zeta_l}\frac{\partial\widetilde{\phi}_1(\xi_l',\zeta_l')}{\partial\zeta_l'} \tag{33}$$

We can calculate the recursive formulation of $\Pi_{\alpha,\beta}^l(\boldsymbol{z}, \boldsymbol{z}')$ and $\Pi_{\beta,\alpha}^l(\boldsymbol{z}, \boldsymbol{z}')$ in the same way, and the random variables $\xi_l, \zeta_l, \xi_l', \zeta_l'$ satisfy Eq.(26).

**Final NTK formulation of complex full-connected network.** Based on the above intermediate limiting kernels, we can calculate the final NTK of complex full-connected networks recursively according to the NTK decomposition Eq. (12). As all widths go to infinity, we can rewrite the NTK decomposition with intermediate kernels as follows.

$$\langle \nabla_{A_l} f(\boldsymbol{z}), \nabla_{A_l} f(\boldsymbol{z}') \rangle = \sigma_A^2 \Pi_\alpha^l(\boldsymbol{z}, \boldsymbol{z}')\Sigma_s^{l-1}(\boldsymbol{z}, \boldsymbol{z}') + \sigma_A^2 \Pi_\beta^l(\boldsymbol{z}, \boldsymbol{z}')\Sigma_r^{l-1}(\boldsymbol{z}, \boldsymbol{z}')$$
$$+ \sigma_A^2 \Pi_{\alpha,\beta}^l(\boldsymbol{z}, \boldsymbol{z}')\Sigma_{s,r}^{l-1}(\boldsymbol{z}, \boldsymbol{z}') + \sigma_A^2 \Pi_{\beta,\alpha}^l(\boldsymbol{z}, \boldsymbol{z}')\Sigma_{r,s}^{l-1}(\boldsymbol{z}, \boldsymbol{z}') \tag{34}$$

$$\langle \nabla_{B_l} f(\boldsymbol{z}), \nabla_{B_l} f(\boldsymbol{z}') \rangle = \sigma_B^2 \Pi_\alpha^l(\boldsymbol{z}, \boldsymbol{z}')\Sigma_r^{l-1}(\boldsymbol{z}, \boldsymbol{z}') + \sigma_B^2 \Pi_\beta^l(\boldsymbol{z}, \boldsymbol{z}')\Sigma_s^{l-1}(\boldsymbol{z}, \boldsymbol{z}')$$
$$- \sigma_B^2 \Pi_{\alpha,\beta}^l(\boldsymbol{z}, \boldsymbol{z}')\Sigma_{r,s}^{l-1}(\boldsymbol{z}, \boldsymbol{z}') - \sigma_B^2 \Pi_{\beta,\alpha}^l(\boldsymbol{z}, \boldsymbol{z}')\Sigma_{s,r}^{l-1}(\boldsymbol{z}, \boldsymbol{z}') \tag{35}$$

$$\langle \nabla_{A_{L+1}} f(\boldsymbol{z}), \nabla_{A_{L+1}} f(\boldsymbol{z}') \rangle + \langle \nabla_{B_{L+1}} f(\boldsymbol{z}), \nabla_{B_{L+1}} f(\boldsymbol{z}') \rangle = \sigma_A^2 \Sigma_s^L + \sigma_B^2 \Sigma_r^L \tag{36}$$

Thus the final NTK of complex full-connected networks can be represented as

$$\widehat{\Theta}(z, z') = \sum_{l=1}^{L} \langle \nabla_{A_l} f(z), \nabla_{A_l} f(z') \rangle + \sum_{l=1}^{L} \langle \nabla_{B_l} f(z), \nabla_{B_l} f(z') \rangle$$

$$= \sum_{l=1}^{L} \left( \Pi_{\alpha}^l(z, z') \Sigma_{\alpha}^l(z, z') + \Pi_{\beta}^l(z, z') \Sigma_{\beta}^l(z, z') \right.$$

$$\left. + \Pi_{\alpha,\beta}^l(z, z') \Sigma_{\alpha,\beta}^l(z, z') + \Pi_{\beta,\alpha}^l(z, z') \Sigma_{\beta,\alpha}^l(z, z') \right) + \Sigma_{\alpha}^{L+1}(z, z') \qquad (37)$$

**Multi-dimensional output.** We denote the above limiting NTK corresponding to the single output complex full-connected network as $\Theta(z, z')$. For multi-dimensional output case, i.e., $f_\theta(z) \in \mathbb{R}^{d_{out}}$, the $i$-th output for $i \in [d_{out}]$ is obtained via

$$[f_\theta(z)]_i = [\Re\{\mathbf{W}_{L+1} h_L\}]_i = \frac{\sigma_A}{\sqrt{n}} a_i^\top s_L - \frac{\sigma_B}{\sqrt{n}} b_i^\top r_L, \qquad (38)$$

where $a_i, b_i \in \mathbb{R}^n$ is the $i$-th row of corresponding real output matrices $A$ and $B$; and $a_i, b_i$ are independent of $a_j, b_j$ for $i \neq j$. As a result, for the multi-dimensional output NTK $\widehat{\mathring{\Theta}}(z, z') \in \mathbb{R}^{d_{out} \times d_{out}}$ we have

$$\left[ \widehat{\mathring{\Theta}}(z, z') \right]_{i,j} = \left\langle \nabla_\theta [f_\theta(z)]_i, \nabla_\theta [f_\theta(z')]_j \right\rangle. \qquad (39)$$

Thus for $i = j$, i.e., the diagonal elements of the kernel matrix, the value satisfies

$$\left[ \mathring{\Theta}(z, z') \right]_{i,i} = \lim_{n \to \infty} \left[ \widehat{\mathring{\Theta}}(z, z') \right]_{i,i} = \Theta(z, z'), \text{ for } i = j; \qquad (40)$$

but for $i \neq j$, since $a_i, b_i$ are independent of $a_j, b_j$ for $i \neq j$ and all the gradient vectors in $\nabla_\theta [f_\theta(x)]_i$ contains $a_i, b_i$ due to the backpropagation, we have

$$\left[ \mathring{\Theta}(z, z') \right]_{i,j} = \lim_{n \to \infty} \left\langle \nabla_\theta [f_\theta(z)]_i, \nabla_\theta [f_\theta(z')]_j \right\rangle = 0, \text{ for } i \neq j. \qquad (41)$$

Therefore, we have the following limiting NTK for the multi-dimensional output complex full-connected network

$$\mathring{\Theta}(z, z') = \Theta(z, z') \otimes \mathbf{I}_{d_{out}}. \qquad (42)$$

This concludes the proof of Theorem 5.

# D   Proof of Theorem 7: Asymptotic equivalence for deep complex networks

To prove Theorem 7, we need to analyze the conditions that the NTK of complex MLP reduces to that of real MLP when $\sigma_A = \sigma_B$. Based on the expansion of final NTK form of complex MLPs (Eq. (34)-(36)), we could conclude that the following four subconditions need to be satisfied at the same time.

**Subcondition 1** $\forall l$, the interaction terms in the NTK decomposition are eliminated.

$$\Pi_{\alpha,\beta}^l(z, z') \Sigma_{s,r}^{l-1}(z, z') + \Pi_{\beta,\alpha}^l(z, z') \Sigma_{r,s}^{l-1}(z, z')$$

$$- \Pi_{\alpha,\beta}^l(z, z') \Sigma_{r,s}^{l-1}(z, z') - \Pi_{\beta,\alpha}^l(z, z') \Sigma_{s,r}^{l-1}(z, z') = 0 \qquad (43)$$

**Subcondition 2** $\forall l$, the interaction terms in $\Pi_{\alpha}^l(z, z')$ and $\Pi_{\beta}^l(z, z')$ are eliminated.

**Subcondition 3** $\forall l$, we have $\Pi_{\alpha}^l(z, z') = \Pi_{\beta}^l(z, z')$.

**Subcondition 4** $\forall l$, we have $\Sigma_s^l(z, z') = \Sigma_r^l(z, z')$.

For Subcondition 1, we can obtain the equivalent formulation as follows

$$\Pi_{\alpha,\beta}^l(z, z') - \Pi_{\beta,\alpha}^l(z, z') = \mathbb{E} Z^{\delta_\alpha^l(z)} Z^{\delta_\beta^l(z')} - \mathbb{E} Z^{\delta_\beta^l(z)} Z^{\delta_\alpha^l(z')} = 0. \qquad (44)$$

because $\Sigma^l_{\alpha,\beta}(z,z') = \Sigma^{l-1}_{s,r}(z,z') - \Sigma^{l-1}_{r,s}(z,z') = 0$ and $\Sigma^l_{\beta,\alpha}(z,z') = 0$ are unachievable. Specifically, for $l = 1$, they hold only when the input $[x,y] = c[x',y']$ for some $c \in \mathbb{R}$. Then we can transform the Subcondition 1 to the following:

$$\mathbb{E}Z^{\boldsymbol{\delta}^l_s(z)\odot\partial\widetilde{\phi}_1(\boldsymbol{\alpha}_l,\boldsymbol{\beta}_l)/\partial\boldsymbol{\alpha}_l+\boldsymbol{\delta}^l_r(z)\odot\partial\widetilde{\phi}_2(\boldsymbol{\alpha}_l,\boldsymbol{\beta}_l)/\partial\boldsymbol{\alpha}_l}Z^{\boldsymbol{\delta}^l_s(z')\odot\partial\widetilde{\phi}_1(\boldsymbol{\alpha}'_l,\boldsymbol{\beta}'_l)/\partial\boldsymbol{\beta}'_l+\boldsymbol{\delta}^l_r(z')\odot\partial\widetilde{\phi}_2(\boldsymbol{\alpha}'_l,\boldsymbol{\beta}'_l)/\partial\boldsymbol{\beta}'_l}$$

$$-\mathbb{E}Z^{\boldsymbol{\delta}^l_s(z)\odot\partial\widetilde{\phi}_1(\boldsymbol{\alpha}_l,\boldsymbol{\beta}_l)/\partial\boldsymbol{\beta}_l+\boldsymbol{\delta}^l_r(z)\odot\partial\widetilde{\phi}_2(\boldsymbol{\alpha}_l,\boldsymbol{\beta}_l)/\partial\boldsymbol{\beta}_l}Z^{\boldsymbol{\delta}^l_s(z')\odot\partial\widetilde{\phi}_1(\boldsymbol{\alpha}'_l,\boldsymbol{\beta}'_l)/\partial\boldsymbol{\alpha}'_l+\boldsymbol{\delta}^l_r(z')\odot\partial\widetilde{\phi}_2(\boldsymbol{\alpha}'_l,\boldsymbol{\beta}'_l)/\partial\boldsymbol{\alpha}'_l} = 0 \tag{45}$$

Based on the Rules. F.13, we can obtain

$$\mathbb{E}Z^{\boldsymbol{\delta}^l_s(z)}Z^{\boldsymbol{\delta}^l_s(z')}\mathbb{E}\frac{\partial\widetilde{\phi}_1(\xi_l,\zeta_l)}{\partial\xi_l}\frac{\partial\widetilde{\phi}_1(\xi'_l,\zeta'_l)}{\partial\zeta'_l} + \mathbb{E}Z^{\boldsymbol{\delta}^l_r(z)}Z^{\boldsymbol{\delta}^l_r(z')}\mathbb{E}\frac{\partial\widetilde{\phi}_2(\xi_l,\zeta_l)}{\partial\xi_l}\frac{\partial\widetilde{\phi}_2(\xi'_l,\zeta'_l)}{\partial\zeta'_l}$$

$$+ \mathbb{E}Z^{\boldsymbol{\delta}^l_s(z)}Z^{\boldsymbol{\delta}^l_r(z')}\mathbb{E}\frac{\partial\widetilde{\phi}_1(\xi_l,\zeta_l)}{\partial\xi_l}\frac{\partial\widetilde{\phi}_2(\xi'_l,\zeta'_l)}{\partial\zeta'_l} + \mathbb{E}Z^{\boldsymbol{\delta}^l_r(z)}Z^{\boldsymbol{\delta}^l_s(z')}\mathbb{E}\frac{\partial\widetilde{\phi}_2(\xi_l,\zeta_l)}{\partial\xi_l}\frac{\partial\widetilde{\phi}_1(\xi'_l,\zeta'_l)}{\partial\zeta'_l}$$

$$- \mathbb{E}Z^{\boldsymbol{\delta}^l_s(z)}Z^{\boldsymbol{\delta}^l_s(z')}\mathbb{E}\frac{\partial\widetilde{\phi}_1(\xi_l,\zeta_l)}{\partial\zeta_l}\frac{\partial\widetilde{\phi}_1(\xi'_l,\zeta'_l)}{\partial\xi'_l} - \mathbb{E}Z^{\boldsymbol{\delta}^l_r(z)}Z^{\boldsymbol{\delta}^l_r(z')}\mathbb{E}\frac{\partial\widetilde{\phi}_2(\xi_l,\zeta_l)}{\partial\zeta_l}\frac{\partial\widetilde{\phi}_2(\xi'_l,\zeta'_l)}{\partial\xi'_l}$$

$$- \mathbb{E}Z^{\boldsymbol{\delta}^l_s(z)}Z^{\boldsymbol{\delta}^l_r(z')}\mathbb{E}\frac{\partial\widetilde{\phi}_1(\xi_l,\zeta_l)}{\partial\zeta_l}\frac{\partial\widetilde{\phi}_2(\xi'_l,\zeta'_l)}{\partial\xi'_l} - \mathbb{E}Z^{\boldsymbol{\delta}^l_r(z)}Z^{\boldsymbol{\delta}^l_s(z')}\mathbb{E}\frac{\partial\widetilde{\phi}_2(\xi_l,\zeta_l)}{\partial\zeta_l}\frac{\partial\widetilde{\phi}_1(\xi'_l,\zeta'_l)}{\partial\xi'_l} = 0 \tag{46}$$

For Subcondition 2, based on Eq. (32) and Eq. (33), we have the expansion as following

$$\mathbb{E}Z^{\boldsymbol{\delta}^l_s(z)}Z^{\boldsymbol{\delta}^l_r(z')}\mathbb{E}\frac{\partial\widetilde{\phi}_1(\xi_l,\zeta_l)}{\partial\xi_l}\frac{\partial\widetilde{\phi}_2(\xi'_l,\zeta'_l)}{\partial\xi'_l} + \mathbb{E}Z^{\boldsymbol{\delta}^l_r(z)}Z^{\boldsymbol{\delta}^l_s(z')}\mathbb{E}\frac{\partial\widetilde{\phi}_2(\xi_l,\zeta_l)}{\partial\xi_l}\frac{\partial\widetilde{\phi}_1(\xi'_l,\zeta'_l)}{\partial\xi'_l}$$

$$+ \mathbb{E}Z^{\boldsymbol{\delta}^l_s(z)}Z^{\boldsymbol{\delta}^l_r(z')}\mathbb{E}\frac{\partial\widetilde{\phi}_1(\xi_l,\zeta_l)}{\partial\zeta_l}\frac{\partial\widetilde{\phi}_2(\xi'_l,\zeta'_l)}{\partial\zeta'_l} + \mathbb{E}Z^{\boldsymbol{\delta}^l_r(z)}Z^{\boldsymbol{\delta}^l_s(z')}\mathbb{E}\frac{\partial\widetilde{\phi}_2(\xi_l,\zeta_l)}{\partial\zeta_l}\frac{\partial\widetilde{\phi}_1(\xi'_l,\zeta'_l)}{\partial\zeta'_l} = 0$$

According to Eq. (29) and Eq. (30), considering $\sigma_A = \sigma_B$, thus $Z^{\boldsymbol{\delta}^l_s(z)}Z^{\boldsymbol{\delta}^l_r(z')} = -Z^{\boldsymbol{\delta}^l_r(z)}Z^{\boldsymbol{\delta}^l_s(z')}$, then we can obtain

$$\mathbb{E}\frac{\partial\widetilde{\phi}_1(\xi_l,\zeta_l)}{\partial\xi_l}\frac{\partial\widetilde{\phi}_2(\xi'_l,\zeta'_l)}{\partial\xi'_l} + \mathbb{E}\frac{\partial\widetilde{\phi}_1(\xi_l,\zeta_l)}{\partial\zeta_l}\frac{\partial\widetilde{\phi}_2(\xi'_l,\zeta'_l)}{\partial\zeta'_l}$$

$$- \mathbb{E}\frac{\partial\widetilde{\phi}_2(\xi_l,\zeta_l)}{\partial\xi_l}\frac{\partial\widetilde{\phi}_1(\xi'_l,\zeta'_l)}{\partial\xi'_l} - \mathbb{E}\frac{\partial\widetilde{\phi}_2(\xi_l,\zeta_l)}{\partial\zeta_l}\frac{\partial\widetilde{\phi}_1(\xi'_l,\zeta'_l)}{\partial\zeta'_l} = 0 \tag{47}$$

For Subcondition 3, given the Subcondition 2 holds, we can obtain that

$$\mathbb{E}\frac{\partial\widetilde{\phi}_1(\xi_l,\zeta_l)}{\partial\xi_l}\frac{\partial\widetilde{\phi}_1(\xi'_l,\zeta'_l)}{\partial\xi'_l} + \mathbb{E}\frac{\partial\widetilde{\phi}_2(\xi_l,\zeta_l)}{\partial\xi_l}\frac{\partial\widetilde{\phi}_2(\xi'_l,\zeta'_l)}{\partial\xi'_l}$$

$$- \mathbb{E}\frac{\partial\widetilde{\phi}_1(\xi_l,\zeta_l)}{\partial\zeta_l}\frac{\partial\widetilde{\phi}_1(\xi'_l,\zeta'_l)}{\partial\zeta'_l} - \mathbb{E}\frac{\partial\widetilde{\phi}_2(\xi_l,\zeta_l)}{\partial\zeta_l}\frac{\partial\widetilde{\phi}_2(\xi'_l,\zeta'_l)}{\partial\zeta'_l} = 0 \tag{48}$$

For Subcondition 4, we can obtain that

$$\mathbb{E}Z^{\boldsymbol{s}_l}Z^{\boldsymbol{s}'_l} - \mathbb{E}Z^{\boldsymbol{r}_l}Z^{\boldsymbol{r}'_l} = \mathbb{E}\widetilde{\phi}_1(\xi_l,\zeta_l)\widetilde{\phi}_1(\xi'_l,\zeta'_l) - \mathbb{E}\widetilde{\phi}_2(\xi_l,\zeta_l)\widetilde{\phi}_2(\xi'_l,\zeta'_l) = 0 \tag{49}$$

where the random variables $\xi_l, \zeta_l, \xi'_l, \zeta'_l$ are distributed as Eq. (26).

It could be verified that there is no common solution for all these four subconditions if we assume that $\partial\widetilde{\phi}_1(\xi_l,\zeta_l)/\partial\xi_l, \partial\widetilde{\phi}_2(\xi_l,\zeta_l)/\partial\xi_l, \partial\widetilde{\phi}_1(\xi_l,\zeta_l)/\partial\zeta_l, \partial\widetilde{\phi}_2(\xi_l,\zeta_l)/\partial\zeta_l$ are all nonzero (this can be verified by substituting all solutions of the Subcondition 1 into the other Subconditions). However, when we allow $\partial\widetilde{\phi}_1(\xi_l,\zeta_l)/\partial\zeta_l, \partial\widetilde{\phi}_2(\xi_l,\zeta_l)/\partial\xi_l$ to be zero, which is common in popular activation functions, we can find the solution that satisfies all the subconditions.

When $\partial\widetilde{\phi}_1(\xi_l,\zeta_l)/\partial\zeta_l = \partial\widetilde{\phi}_2(\xi_l,\zeta_l)/\partial\xi_l = 0$ and $\partial\widetilde{\phi}_1(\xi_l,\zeta_l)/\partial\xi_l = \partial\widetilde{\phi}_2(\xi_l,\zeta_l)/\partial\zeta_l$, the Subcondition 2 and Subcondition 3 naturally hold. For Subcondition 1, it is transformed to

$$\mathbb{E}Z^{\boldsymbol{\delta}^l_s(z)}Z^{\boldsymbol{\delta}^l_r(z')}\mathbb{E}\frac{\partial\widetilde{\phi}_1(\xi_l,\zeta_l)}{\partial\xi_l}\frac{\partial\widetilde{\phi}_2(\xi'_l,\zeta'_l)}{\partial\zeta'_l} - \mathbb{E}Z^{\boldsymbol{\delta}^l_r(z)}Z^{\boldsymbol{\delta}^l_s(z')}\mathbb{E}\frac{\partial\widetilde{\phi}_2(\xi_l,\zeta_l)}{\partial\zeta_l}\frac{\partial\widetilde{\phi}_1(\xi'_l,\zeta'_l)}{\partial\xi'_l} = 0$$

which needs $\mathbb{E}Z^{\delta_s^l(z)}Z^{\delta_r^l(z')} - \mathbb{E}Z^{\delta_r^l(z)}Z^{\delta_s^l(z')} = 0$, and this is only satisfied when $\Pi_{\alpha,\beta}^l(z,z') = \Pi_{\beta,\alpha}^l(z,z')$, thus this constructs a recursion: $\mathbb{E}Z^{\delta_s^l(z)}Z^{\delta_r^l(z')} - \mathbb{E}Z^{\delta_r^l(z)}Z^{\delta_s^l(z')} = 0$ only when $\mathbb{E}Z^{\delta_s^{l+1}(z)}Z^{\delta_r^{l+1}(z')} - \mathbb{E}Z^{\delta_r^{l+1}(z)}Z^{\delta_s^{l+1}(z')} = 0$. Finally, due to that for the output layer, $Z^{\delta_s^L(z)}$ and $Z^{\delta_r^L(z')}$ are independent, i.e., $\mathbb{E}Z^{\delta_s^L(z)}Z^{\delta_r^L(z')} = \mathbb{E}Z^{\delta_r^L(z)}Z^{\delta_s^L(z')} = 0$, thus this concludes the proof of satisfying Subcondition 1.

For Subcondition 4, when $\widetilde{\phi}_2(a,b) = \widetilde{\phi}_1(b,-a)$, which could also induce that $\partial\widetilde{\phi}_1(\xi_l,\zeta_l)/\partial\xi_l = \partial\widetilde{\phi}_2(\xi_l,\zeta_l)/\partial\zeta_l$, we can obtain

$$\mathbb{E}\widetilde{\phi}_2(\xi_l,\zeta_l)\widetilde{\phi}_2(\xi_l',\zeta_l') = \mathbb{E}\widetilde{\phi}_1(\zeta_l,-\xi_l)\widetilde{\phi}_1(\zeta_l',-\xi_l') \tag{50}$$

Notice that

$$(\zeta_l,-\xi_l,\zeta_l',-\xi_l') \sim \mathcal{N}\left(0, \begin{pmatrix} \Sigma_\alpha^l(z,z) & -\Sigma_{\beta,\alpha}^l(z,z) & \Sigma_\alpha^l(z,z') & -\Sigma_{\beta,\alpha}^l(z,z') \\ -\Sigma_{\alpha,\beta}^l(z,z) & \Sigma_\beta^l(z,z) & -\Sigma_{\alpha,\beta}^l(z,z') & \Sigma_\beta^l(z,z') \\ \Sigma_\alpha^l(z',z) & -\Sigma_{\beta,\alpha}^l(z',z) & \Sigma_\alpha^l(z',z') & -\Sigma_{\beta,\alpha}^l(z',z') \\ -\Sigma_{\alpha,\beta}^l(z',z) & \Sigma_\beta^l(z',z) & -\Sigma_{\alpha,\beta}^l(z',z') & \Sigma_\beta^l(z',z') \end{pmatrix}\right) \tag{51}$$

due to that $\Sigma_{\alpha,\beta}^l(z',z) = -\Sigma_{\beta,\alpha}^l(z,z')$, thus the distribution is the same as Eq.(26). Then we can obtain that

$$\mathbb{E}\widetilde{\phi}_2(\xi_l,\zeta_l)\widetilde{\phi}_2(\xi_l',\zeta_l') = \mathbb{E}\widetilde{\phi}_1(\zeta_l,-\xi_l)\widetilde{\phi}_1(\zeta_l',-\xi_l') = \mathbb{E}\widetilde{\phi}_1(\xi_l,\zeta_l)\widetilde{\phi}_1(\xi_l',\zeta_l') \tag{52}$$

In summary, when $\widetilde{\phi}_2(a,b) = \widetilde{\phi}_1(b,-a)$ and $\partial\widetilde{\phi}_1(\xi_l,\zeta_l)/\partial\zeta_l = \partial\widetilde{\phi}_2(\xi_l,\zeta_l)/\partial\xi_l = 0$ the NTK of complex MLP reduces to real MLP.

As for the necessary and sufficient conditions for the asymptotic equivalence, we consider all solutions of Subcondition 4: $\widetilde{\phi}_2(a,b) = \widetilde{\phi}_1(-b,a)$ and $\widetilde{\phi}_2(a,b) = \widetilde{\phi}_1(b,-a)$, and verify them with Subcondition 1-3. We can obtain that all solutions satisfying all the conditions as follows $\widetilde{\phi}_2(a,b) = \widetilde{\phi}_1(-b,a)$.

$$\frac{\partial\widetilde{\phi}_1(\xi_l,\zeta_l)}{\partial\zeta_l} = \frac{\partial\widetilde{\phi}_2(\xi_l,\zeta_l)}{\partial\xi_l} = 0, \quad \frac{\partial\widetilde{\phi}_1(\xi_l,\zeta_l)}{\partial\xi_l} = \pm\frac{\partial\widetilde{\phi}_2(\xi_l,\zeta_l)}{\partial\zeta_l}; \tag{53}$$

$$\frac{\partial\widetilde{\phi}_1(\xi_l,\zeta_l)}{\partial\xi_l} = \frac{\partial\widetilde{\phi}_2(\xi_l,\zeta_l)}{\partial\zeta_l} = 0, \quad \frac{\partial\widetilde{\phi}_1(\xi_l,\zeta_l)}{\partial\zeta_l} = \pm\frac{\partial\widetilde{\phi}_2(\xi_l,\zeta_l)}{\partial\xi_l}. \tag{54}$$

This concludes the proof of the Condition 2 in Theorem 7.

For split activation functions, i.e.,

$$\phi(z) = \phi_1(z) + \phi_2(z)i = \phi_R(\Re(z)) + \phi_R(\Im(z))i$$

we naturally have $\partial\widetilde{\phi}_1(\xi_l,\zeta_l)/\partial\zeta_l = \partial\widetilde{\phi}_2(\xi_l,\zeta_l)/\partial\xi_l = 0$ and $\partial\widetilde{\phi}_1(\xi_l,\zeta_l)/\partial\xi_l = \partial\widetilde{\phi}_2(\xi_l,\zeta_l)/\partial\zeta_l$. Thus Subcondition 1-3 hold as analyzed above. For Subcondition 4, we have $\mathbb{E}Z^{s_l}Z^{s_l'} = \mathbb{E}\widetilde{\phi}_1(\xi)\widetilde{\phi}_1(\xi')$ and $\mathbb{E}Z^{r_l}Z^{r_l'} = \mathbb{E}\widetilde{\phi}_2(\zeta)\widetilde{\phi}_2(\zeta')$, where

$$(\xi,\xi') \sim \mathcal{N}\left(0, \begin{pmatrix} \Sigma_\alpha^l(z,z) & \Sigma_\alpha^l(z,z') \\ \Sigma_\alpha^l(z,z') & \Sigma_\alpha^l(z',z') \end{pmatrix}\right), \tag{55}$$

$$(\zeta,\zeta') \sim \mathcal{N}\left(0, \begin{pmatrix} \Sigma_\beta^l(z,z) & \Sigma_\beta^l(z,z') \\ \Sigma_\beta^l(z,z') & \Sigma_\beta^l(z',z') \end{pmatrix}\right). \tag{56}$$

Note that $\Sigma_\alpha^l(z,z') = \Sigma_\beta^l(z,z')$ for $\forall z,z' \in \mathbb{C}^d$ since $\sigma_A = \sigma_B$. This concludes the proof of the Condition 1 in Theorem 7.

## E  Additional experiments for complex NTK during training

**Verifying complex NTK during training.** In this experiment, we investigate the difference of complex NTK before and after training on MNIST, i.e., we compare the empirical complex NTKs

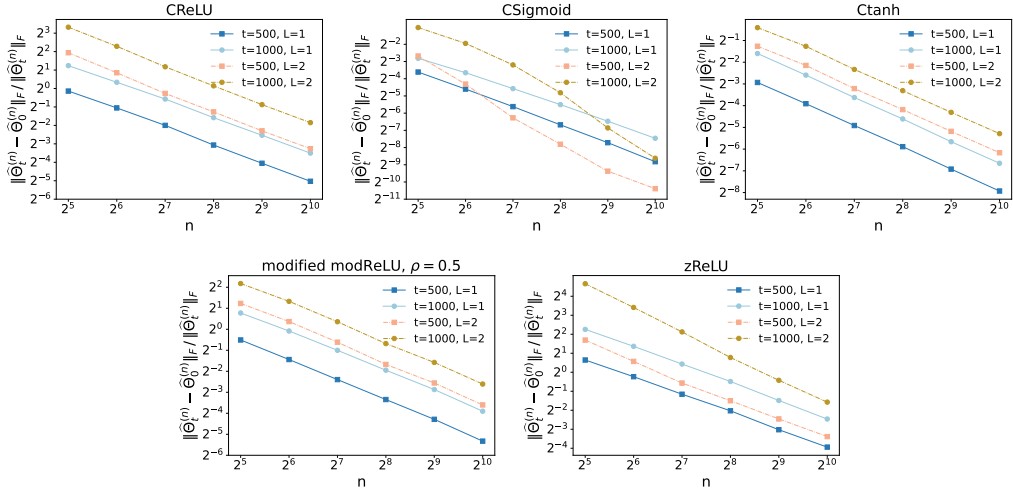

Figure 4: **Convergence of the difference of empirical complex-valued NTKs before and during training as widths grow.** Similar to Figure 3, but the Y-axis is the relative Frobenius norm of the change of empirical NTKs during training.

during training $\hat{\Theta}_t^{(n)}$ with empirical complex NTK at initialization $\hat{\Theta}_0^{(n)}$. The results are shown in Figure 4. We observe that the relative norm converges fast in all these cases, regardless of the training steps, hidden layer number and activation functions. So the empirical complex NTK during training $\hat{\Theta}_t^{(n)}$ converges to the empirical complex NTK at initialization $\hat{\Theta}_0^{(n)}$ as the widths grows, which means the complex NTK remains frozen during training. This experiment perfectly verifies our Theorem 4, because all these activation functions satisfy the the Assumption 2 of complex tensor program.

In conclusion, the first two experiments focus on the difference between complex NTK and real NTK at initialization, which verify our theoretical results about real-valued backpropagation and demonstrate that our conditions in Theorem 7 are non-vacuous. And this experiment focuses on the change of NTKs during training, which verifies our Theorem 4; and on the other hand, it indicates that in order to figure out the relationship between complex NTKs and real NTKs, we could just focus on the NTKs at initialization.

# F NETSOR$^\top$ Program

For self-containedness, in this section we will highlight some important theoretical tools used in our proofs.

For MLPs and CNNs, it has been shown that the pre-activation of each layer tends to a Gaussian process as width $n \to \infty$ based on the Central Limit Theorem (CLT) [Lee et al., 2018] and the neural tangent kernel (NTK) at initialization tends to a limiting kernel as width $n \to \infty$ [Jacot et al., 2018, Arora et al., 2019]. However, the analysis can not be extended to the neural networks with weight sharing, like recurrent neural networks (RNN), because the sequential limit is not possible and conditioning on previous layers does not result in iid weights. To deal with it, Yang [2019b, 2020] presented a novel proof using Gaussian conditioning trick which allows the recurrent weights in a network. Further, it has been demonstrated that any architecture that can be expressed as NETSOR programs converges to Gaussian process as all widths go to infinity in the same rate and any architecture whose feed-forward and backward procedure can be expressed as NETSOR$^\top$ programs has a convergent NTK as all widths go to infinity in the same rate. Besides, to ensure readability, we did not use the original Tensor Programs [Yang, 2019a], which has been superceded by the NETSOR$^\top$, a more concise and readable version. We list the important theoretical tools used in our proofs including definitions, theorems and conditions from [Yang, 2019b, 2020] as follows.

**Definition F.11 (Simplified NETSOR$^\top$ Program)** *A NETSOR$^\top$ program is just a sequence of $\mathbb{R}^n$ vectors inductively generated via one of the following ways from an initial set $\mathcal{V}$ of random $\mathbb{R}^n$ vectors and a set $\mathcal{W}$ of random $n \times n$ matrices*

**Nonlin** *Given $\phi: \mathbb{R}^k \to \mathbb{R}$ and $\boldsymbol{x}^1, \ldots, \boldsymbol{x}^k \in \mathbb{R}^n$, we can generate $\phi(\boldsymbol{x}^1, \ldots, \boldsymbol{x}^k) \in \mathbb{R}^n$.*

**MatMul** *Given $\boldsymbol{W} \in \mathbb{R}^{n \times n}$ and $\boldsymbol{x} \in \mathbb{R}^n$, we can generate $\boldsymbol{W}\boldsymbol{x} \in \mathbb{R}^n$ or $\boldsymbol{W}^\top \boldsymbol{x} \in \mathbb{R}^n$.*

**Condition F.12 (Simple GIA Check)** *The output layer is sampled independently and with zero mean from all other parameters and is not used anywhere else in the interior of the network.*

**Rules F.13 (Intuitions for Computing the Limits)** *When the width $n \gg 1$, every (pre-)actication vector $\boldsymbol{x} \in \mathbb{R}^n$ has roughly i.i.d. coordinates distributed as some random variable denoted $Z^{\boldsymbol{x}}$. Thus for any vector $\boldsymbol{x}, \boldsymbol{y} \in \mathbb{R}^n$, as $n \to \infty$,*

$$\boldsymbol{x}^\top \boldsymbol{y}/n = \frac{1}{n} \sum_{i=1}^n \boldsymbol{x}_i \boldsymbol{y}_i \to \mathbb{E} Z^{\boldsymbol{x}} Z^{\boldsymbol{y}},$$

*Then we can use the following rules to compute $Z^{\boldsymbol{x}}$ of a NETSOR$^\top$ program recursively.*

- **Nonlin** *For any fixed $k$ and $\phi: \mathbb{R}_k \to \mathbb{R}$, we have*

$$Z^{\phi(\boldsymbol{x}_1, \cdots, \boldsymbol{x}_k)} = \phi(Z^{\boldsymbol{x}_1}, \cdots, Z^{\boldsymbol{x}_k})$$

- **MatMul** *For any set of $\mathbb{R}^n$ vectors $\mathcal{X}$ and a matrix $\mathbf{W} \in \mathbb{R}^{n \times n}$ with $\mathbf{W}_{ij} \sim \mathcal{N}(0, \sigma_W^2/n)$, the set of random variables $\{Z^{\mathbf{W}\boldsymbol{x}} : \boldsymbol{x} \in \mathcal{X}\}$ is jointly Gaussian with zero mean and covariance*

$$Cov(Z^{\mathbf{W}\boldsymbol{x}}, Z^{\mathbf{W}\overline{\boldsymbol{x}}}) = \sigma_W^2 \mathbb{E} Z^{\boldsymbol{x}} Z^{\overline{\boldsymbol{x}}}, \quad \forall \boldsymbol{x}, \overline{\boldsymbol{x}} \in \mathcal{X}.$$

  *If $\mathcal{Y}$ is any set of $\mathbb{R}^n$ vectors and $\overline{\mathbf{W}} \neq \mathbf{W}$, then $\{Z^{\mathbf{W}\boldsymbol{x}} : \boldsymbol{x} \in \mathcal{X}\}$ is independent from $\{Z^{\overline{\mathbf{W}}\boldsymbol{y}} : \boldsymbol{y} \in \mathcal{Y}\}$.*

**Theorem F.14 (NETSOR$^\top$ Master Theorem)** *Consider a NETSOR$^\top$ program. Suppose:*

- *for each intial $\mathbf{W} \in \mathcal{W}$, $\mathbf{W}_{ij} \in \mathcal{N}(0, \sigma_W^2/n)$ for an associate variance $\sigma_W^2$;*

- *there is a multivariate Gaussian $Z^{\mathcal{V}} = \{Z^{\boldsymbol{\nu}} : \boldsymbol{\nu} \in \mathcal{V} \in \mathbb{R}^{|\mathcal{V}|}\}$ such that the initial set of vectors $\mathcal{V}$ are sampled like $\{\boldsymbol{\nu}_i : \boldsymbol{\nu} \in \mathcal{V}\} \sim Z^{\mathcal{V}}$ i.i.d. for each $i \in [n]$.*

*if the program satisfies the simple GIA check and all $\phi$ used in **Nonlin** are polynomially bounded, then*

$$\frac{1}{n} \sum_{i=1}^n \psi\left(h_i^1, \ldots, h_i^k\right) \overset{a.s.}{\to} \mathbb{E}\psi\left(Z^{h^1}, \ldots, Z^{h^k}\right), \quad as \quad n \to \infty$$

*for any collection of vectors $h^1, \ldots, h^k$ in the program and any polynomially bounded $\psi: \mathbb{R}^k \to \mathbb{R}$, where $Z^{h^i}$ are defined in **Rules F.13**.*