# OpenReview forum: "Real-Valued Backpropagation is Unsuitable for Complex-Valued Neural Networks"
_NeurIPS.cc/2022/Conference — NeurIPS 2022 Accept_

### Official Review · Reviewer_E55z · 2022-07-01

**Rating:** 8
**Confidence:** 4
**Soundness:** 4 excellent
**Presentation:** 3 good
**Contribution:** 3 good

**Summary:**

The paper derives the equation for the neural tangent kernel (NTK) for complex-valued neural networks trained via real-valued backpropagation (backpropagation on the real and imaginary components separately). Their main theoretical result shows that, under common choices of activation functions, the NTK in the infinite-width limit is the same as that of a real-valued neural network.

**Questions:**

- Please provide a proofreading because the paper has several typos.
- A small introduction to questions of holomorphicity or complex backpropagation could be helpful to broaden the understanding of the results.

**Strengths And Weaknesses:**

This is an extremely interesting paper for the community working on complex-valued neural networks. The results are easy to follow and, by any point of view, extremely interesting.

---

> ### Author Response · Authors · 2022-08-02
> **Many thanks for the constructive reviews!**
>
> Many thanks for the constructive comments! We sincerely appreciate that you acknowledge our contributions.
>
> We will fix the typos in the revised version. Holomorphicity and complex backpropagation were introduced in the appendix, we will revise our paper and add an introduction to them in the main text according to the reviews.
>
> Best Regards,
>
> Authors

---

### Official Review · Reviewer_JGv9 · 2022-07-11

**Rating:** 5
**Confidence:** 3
**Soundness:** 3 good
**Presentation:** 4 excellent
**Contribution:** 3 good

**Summary:**

Complex-valued infinitely wide neural networks are considered in the paper. Through studying the kernel limit of complex-valued networks, the authors showed that real-valued backpropagation reduces complex NTK to a real NTK, under certain conditions on the activation function. Interestingly, many of practically used activation functions satisfy the condition, and therefore, using real-valued backpropagation may not be suitable to leverage all the potential of complex-valued networks.

**Questions:**

I am interested to know if complex-valued neural networks can beat the state-of-the-art real-valued neural networks in applications, such as computer vision and time series prediction.

Is there any concrete motivation of studying the infinite width limit of complex-valued neural networks? As mentioned in the introduction, complex-valued neural networks have richer representation capacity and faster learning. Is overparameterization in complex-valued neural networks necessary for a good performance?

Is it possible to suggest a proper backpropagation method based on the theoretical findings in the paper in order to further leverage the potential of complex-valued neural networks?

**Limitations:**

The major limitation comes from the NTK analysis. In practice, neural networks are often trained with a relatively large learning rate for a sufficiently large number of iterations. Therefore, the working regime of neural networks is actually not NTK. The theoretical findings undoubtedly enhance the understanding of complex-valued networks; they may have limited implications for practice. For example, in the infinite-width limit, real-valued backpropagation may not be appropriate. However, in practice, the theory does not apply due to aforementioned reasons. Whether using real-valued backpropagation is suitable remains open.

I am fond of the theoretical findings in the paper. But the practical value of the paper is rather limited to my knowledge. Therefore, I have a neutral initial rating of the paper.

**Strengths And Weaknesses:**

The theoretical findings in the paper are interesting and intriguing. Complex-valued neural networks are relatively less studied, and whose theoretical understandings are lacking. This paper contributes towards this direction and the reported results read sound and correct. Moreover, there are empirical studies corroborate the theoretical findings. The organization of the paper is good and most of the results are easy to understand.

I will discuss some weaknesses of the paper in the limitation section.

---

> ### Author Response · Authors · 2022-08-02
> **Many thanks for the constructive reviews!**
>
> Many thanks for the constructive reviews! We sincerely appreciate that you acknowledge our contributions and theoretical findings. We will improve our writing and address the concerns.
>
> ---
>
> **Q1: I am interested to know if complex-valued neural networks can beat the state-of-the-art real-valued neural networks in applications, such as computer vision and time series prediction.**
>
> A1: Compared to real-valued neural networks (RVNNs), complex-valued neural networks (CVNNs) are relatively less studied, but CVNNs have found many applications, especially in signal processing. For example, the CVNNs outperform state-of-the-art RVNNs in many polarimetric SAR image classification tasks due to exploiting the phase information in SAR image [3]; achieve state-of-the-art performance on some audio-related tasks [2] and several tasks involving very long-term dependencies [1]. Besides, various applications of CVNNs are summarized in [4].
>
> [1]. Martin Arjovsky, et al. Unitary evolution recurrent neural networks. ICML 2016.
>
> [2]. Chiheb Trabelsi, et al. Deep Complex Networks. ICLR 2018.
>
> [3]. Zhimian Zhang, et al. Complex-valued convolutional neural network and its application in polarimetric SAR image classification. IEEE Transactions on Geoscience and Remote Sensing 2017.
>
> [4]. Joshua Bassey, et al. A survey of complex-valued neural networks. arXiv 2021.
>
> ---
>
> **Q2: Is there any concrete motivation of studying the infinite width limit of complex-valued neural networks? As mentioned in the introduction, complex-valued neural networks have richer representation capacity and faster learning. Is overparameterization in complex-valued neural networks necessary for a good performance?**
>
> A2: First, about the motivation of the study: In the field of complex-valued neural networks (CVNNs), one of the most basic problems is to study the fundamental difference between CVNNs and real-valued neural networks (RVNNs). Using backpropagation, could CVNNs learn different hypotheses with RVNNs? It is very difficult to answer. Over the past thirty years, due to limitations of technical tools and understanding of deep neural networks, despite many empirical studies, it is hard, or even intractable, to analyze the training dynamics of neural networks theoretically.
>
> Second, about the overparameterization: By generalizing NTK theory to the complex domain, we give a theoretical answer to this fundamental question in an asymptotic sense (infinite width). We prove the asymptotic equivalence between CVNNs and RVNNs using real-valued BP under most commonly used activation functions. Furthermore, although the results are on infinitely wide networks, experiments show that this equivalence is already evident at common widths and verify the effectiveness of our theoretical results.
>
> Overall, we claim that in this paper, NTK theory is only a theoretical tool, based on which we find a way to solve the above fundamental problem in the field of CVNNs. And the obtained results are verified empirically to be effective at common widths. In the future, we believe there will be more powerful techniques for a better analysis of non-overparameterized networks, but this is beyond the scope of this paper.
>
> ---
>
> **Q3: Is it possible to suggest a proper backpropagation method based on the theoretical findings in the paper in order to further leverage the potential of complex-valued neural networks?**
>
> A3: Based on our results, the backpropagation method with complex differentials, or backpropagation method based on wirtinger calculas have the potential to leverage CVNNs.
>
> ---
>
> **Q4: The major limitation comes from the NTK analysis. In practice,... they may have limited implications for practice.**
>
> A4: First, despite analysis on infinitely wide networks, experiments demonstrate that our theoretical results are already evident at common widths. Second, recent studies have found many applications for NTKs, and show that even for advanced deep architectures, the corresponding NTKs could help a lot in dataset distillation task [5], hyperparameter transfer for large models [6], in which impressing results based on NTK are obtained.
>
> Besides, as claimed above, for this paper, NTK theory is the most powerful existing theoretical tool for analyzing training dynamics, based on which we firstly provide a way to solve the fundamental problem in the field of CVNNs. The results could guide the design of CVNNs in practice. In the future, we believe there will be more powerful techniques for a better analysis of non-overparameterized networks, but this is beyond the scope of this paper.
>
> [5]. Timothy Nguyen, et al. Dataset distillation with infinitely wide convolutional networks. NeurIPS 2021.
>
> [6]. Greg Yang, et al. Tuning large neural networks via zero-shot hyperparameter transfer. NeurIPS 2021.
>
> ---
>
> We are happy to respond further if there is something unclear. We appreciate the thoughtful questions, and we will revise our paper according to the constructive reviews.

---

> > ### Comment · Reviewer_JGv9 · 2022-08-09
> > **Thank you for the response**
> >
> > Thanks for the detailed response to each of my questions and comments.
> >
> > A small comment on NTK: Despite successes of NTK approximation of neural networks [5, 6], many other results show that NTKs cannot give rise to comparable performance in practice. The major drawback of NTK approximation is it assumes the parameters in the network only move very little from their initialization.
> >
> > Overall, the authors addressed most of my concerns (practical implications are a bit weak but do not undermine the interesting theoretical findings). I am willing to keep my positive rating.

---

> > > ### Author Response · Authors · 2022-08-09
> > > **Many thanks for your kind reply!**
> > >
> > > Dear Reviewer JGv9,
> > >
> > > Many thanks for your kind reply, acknowledging our contribution, and voting for the acceptance of our submission. We will revise our paper according to the suggestions.
> > >
> > > ---
> > >
> > > Answer: Thank you very much for your feedback.
> > >
> > > Even if NTKs have some successful practical applications (such as in [5], which used the NTKs of CNNs to get impressing distillation results that even neural networks can't achieve), we also admit that for classic classification problems, there are still some gaps between NTKs and the best results after tuning in advanced network architectures.
> > >
> > > However, what we want to claim is that, the main contribution of our paper is turning an otherwise intractable fundamental question in CVNNs into one that could be answered and identified, in the asymptotic sense with NTK theoretical tools. The theoretical analysis of this problem was not possible before. As a result, we obtained some surprising findings and numerically verified the findings. Note that the real-valued backpropagation method has been the most common method for optimizing CVNNs in the past three decades. Thus despite the limitation of the tool used (NTK), we believe we've taken a step forward on this fundamental theoretical problem.
> > >
> > > ---
> > >
> > > Best Regards,
> > >
> > > Authors

---

### Official Review · Reviewer_WKtx · 2022-07-11

**Rating:** 8
**Confidence:** 5
**Soundness:** 4 excellent
**Presentation:** 4 excellent
**Contribution:** 3 good

**Summary:**

This paper extend the Tensor programs that was initially proposed for real valued neural networks to complex valued neural networks (CVNN), and as a result, providing the foundations to calculate the the neural tangent kernel (NTK) of any CVNN architecture. Later, they show that the NTK of any CVNN with a certain type of activation functions ( that contains majority of common used activation function ) is as expressive a real valued neural network by comparing their limiting NTKs, as a result CVNNs can not offer anything more that real valued neural networks. They also provide some examples of activation functions that differs CVNNs from their real valued neural networks.

**Questions:**

How would this results be affected if we replace the output layer instead of Real(WH), we use abs(WH)? can tensor programs still be applied to these kind of networks? how does it affect the results?

**Ethics Review Area:**

["I don’t know"]

**Limitations:**

It would be very useful to see the of effects of choosing different activation functions on performance of CVNNs and real neural works in real world datasets. How can zrelu be superior over other cost functions? What are the task that CVNNs are better than their exact real valued counterpart? A few carefully experiments on real valued neural networks can to answer these question can significantly boost the quality of this paper.

**Strengths And Weaknesses:**

Tensor programs is a series of papers that build the foundations for understanding wide neural network. It was initially proposed for real valued neural network to calculate the NTK of any architecture. This work extend it to CVNNs and thus provide a theoretical foundation to analysis CVNNs. The results of this theoretical foundation, other than calculating the NTK of any CVNN can be very valuable, one that is discussed in this paper on conditions on activation functions that distinguishes CVNNs from their real valued neural networks. The foundation that is built in this work also has the potential to be used to extend the theorems to analyze CVNNs outside of NTK regime where weights move from initialization like Tensor plus programs [1].



[1] Yang, Greg, and Edward J. Hu. "Feature learning in infinite-width neural networks." arXiv preprint arXiv:2011.14522 (2020).

---

> ### Author Response · Authors · 2022-08-02
> **Many thanks for the constructive reviews!**
>
> Many thanks for the constructive reviews! We appreciate that you acknowledge our contributions.
>
> ---
>
> Q1: How would this results be affected if we replace the output layer instead of $\Re$ $(\mathbf{Wh})$, we use $|\mathbf{Wh}|$? can tensor programs still be applied to these kind of networks? how does it affect the results?
>
> A1: Thank you for the insightful question. Replacing the output layer with $|\mathbf{Wh}|$ will not affect our results. The reasons are that, first, as written in Section 2.1, our analysis can be naturally applied to complex-valued output by decomposing the real and imaginary part of output into two functions; second, the NTK captures the internal training behavior of network structures. Specifically, the NTK quantifies how much the function changes $\Delta f$ in function space as we take a small gradient step in parameter space $\Delta \theta$; and the cost functions do not affect the NTKs. As a result, the 'abs' operator could be seen as in cost functions, and a normal network structure with a complex-valued linear output layer is still used, and thus its NTK is unchanged.
>
> ---
>
> Thank you for the thoughtful suggestions. We will further conduct more experiments to evaluate different activation functions, and improve our paper accordingly.

---

### Meta-Review · Area_Chair_G3mv · 2022-08-28

**Recommendation:** Accept
**Confidence:** Less certain

**Metareview:**

This paper extends Yang's Tensor Program paradigm to the setting of complex-valued neural networks and shows that under certain conditions the infinite-width training dynamics under real-valued backpropagation are equivalent to those of real-valued networks. Numerical evidence supporting the main conclusions is presented and a short discussion argues for the necessity of backpropagation algorithms specifically designed for complex networks.

Even though the idea is not entirely novel, the proof that real-valued backpropagation is unsuitable for complex-valued neural networks will likely surprise many readers, and it is useful to present a solid theoretical explanation, even if it only holds in a certain infinite-width regime. The extension of Tensor Programs to complex networks is also a nice result in itself, if not groundbreaking. The reviewers generally appreciate these strengths of the paper.

The paper would be more impactful if further effort were devoted to the practical implications of the results, including a direct comparison with a backpropagation method designed for complex networks. It would be particularly useful to characterize the types of functions that are more readily learned with a method based on, e.g., Wirtinger calculus.

In light of the the above strengths and weaknesses, this is a borderline paper. Ultimately, I believe it falls just above the threshold and I recommend acceptance.

**Award:**

No

---

### Decision · Program_Chairs · 2022-09-14

Accept